# A Unified Perspective on Natural Gradient Variational Inference with Gaussian Mixture Models

**Oleg Arenz**                                               *oleg.arenz@tu-darmstadt.de*
*Intelligent Autonomous Systems*
*Technical University of Darmstadt*

**Philipp Dahlinger**                                        *philipp.dahlinger@kit.edu*
*Autonomous Learning Robots*
*Karlsruhe Institute of Technology*

**Zihan Ye**                                                 *zihan.ye@tu-darmstadt.de*
*Artificial Intelligence & Machine Learning*
*Hessian Center for AI (hessian.AI)*
*Technical University of Darmstadt*

**Michael Volpp**                                            *michael.volpp@kit.edu*
**Gerhard Neumann**                                          *gerhard.neumann@kit.edu*
*Autonomous Learning Robots*
*Karlsruhe Institute of Technology*

**Reviewed on OpenReview:** *https://openreview.net/forum?id=tLBjsX4tjs*

## Abstract

Variational inference with Gaussian mixture models (GMMs) enables learning of highly tractable yet multi-modal approximations of intractable target distributions with up to a few hundred dimensions. The two currently most effective methods for GMM-based variational inference, VIPS and iBayes-GMM, both employ independent natural gradient updates for the individual components and their weights. We show for the first time, that their derived updates are equivalent, although their practical implementations and theoretical guarantees differ. We identify several design choices that distinguish both approaches, namely with respect to sample selection, natural gradient estimation, stepsize adaptation, and whether trust regions are enforced or the number of components adapted. We argue that for both approaches, the quality of the learned approximations can heavily suffer from the respective design choices: By updating the individual components using samples from the mixture model, iBayes-GMM often fails to produce meaningful updates to low-weight components, and by using a zero-order method for estimating the natural gradient, VIPS scales badly to higher-dimensional problems. Furthermore, we show that information-geometric trust-regions (used by VIPS) are effective even when using first-order natural gradient estimates, and often outperform the improved Bayesian learning rule (iBLR) update used by iBayes-GMM. We systematically evaluate the effects of design choices and show that a hybrid approach significantly outperforms both prior works. Along with this work, we publish our highly modular and efficient implementation for natural gradient variational inference with Gaussian mixture models, which supports 432 different combinations of design choices, facilitates the reproduction of all our experiments, and may prove valuable for the practitioner.

# 1 Introduction

Many problems in machine learning involve inference from intractable distributions $p(\mathbf{x})$, that might further only be known up to a normalizing constant $Z$, that is, $p(\mathbf{x}) = \frac{1}{Z}\tilde{p}(\mathbf{x})$. For example when learning latent variable models, $\tilde{p}(\mathbf{x})$ corresponds to the intractable distribution over the latent variable (Volpp et al., 2023); and in maximum entropy reinforcement learning, $\tilde{p}(\mathbf{x})$ corresponds to the exponentiated return, $\tilde{p}(\mathbf{x}) = \exp R(x)$ of trajectory $\mathbf{x}$ (Ziebart, 2010). Bayesian inference is another example, where the intractable, unnormalized target distribution $\tilde{p}(\mathbf{x})$ corresponds to the product of prior and likelihood. However, whereas in Bayesian inference, there is particular interest in scaling to high-dimensional problems and large datasets, we stress that our work considers problems of moderate dimensionality of up to a few hundred of dimensions, where modeling the full covariance matrix of Gaussian distributions is still tractable. Important applications can be found, for example, in robotics—where $p(\mathbf{x})$ could be a multimodal distribution over joint-configurations that reach the desired pose (Pignat et al., 2020) or over collision-free motions that reach a given goal (Ewerton et al., 2020)—, or in non-amortized variational inference for latent variable models (Volpp et al., 2023).

Variational inference (VI) aims to approximate the intractable target distribution $\tilde{p}(\mathbf{x})$ by means of a tractable, parametric model $\tilde{q}_{\boldsymbol{\theta}}(\mathbf{x})$, with parameters $\boldsymbol{\theta}$. Variational inference is typically framed as the problem of maximizing the evidence lower bound objective (ELBO), which is equivalent to minimizing the reverse Kullback-Leibler (KL) divergence (Kullback and Leibler, 1951), $\mathrm{KL}(q_{\boldsymbol{\theta}}||p)$, between approximation $q_{\boldsymbol{\theta}}$ and target distribution $p$. The reverse KL divergence is a principled choice for the optimization problem, as it directly corresponds to the average amount of information (measured in nats), that samples from the approximation $q_{\boldsymbol{\theta}}(\mathbf{x})$ contain for discriminating between the approximation and the target distribution. In this work, we focus on a particular choice of variational distribution—a Gaussian mixture model $q_{\boldsymbol{\theta}}(\mathbf{x}) = \sum_o q_{\boldsymbol{\theta}}(o)q_{\boldsymbol{\theta}}(\mathbf{x}|o)$ with weights $q_{\boldsymbol{\theta}}(o)$ and Gaussian components $q_{\boldsymbol{\theta}}(\mathbf{x}|o)$. The ELBO is then given by

$$J(\boldsymbol{\theta}) = \sum_o q_{\boldsymbol{\theta}}(o) \int_{\mathbf{x}} q_{\boldsymbol{\theta}}(\mathbf{x}|o) \log \tilde{p}(\mathbf{x})d\mathbf{x} + H(q_{\boldsymbol{\theta}}), \tag{1}$$

where $H(q_{\boldsymbol{\theta}}) = -\int_{\mathbf{x}} q_{\boldsymbol{\theta}}(\mathbf{x})\log(q_{\boldsymbol{\theta}}(\mathbf{x}))d\mathbf{x}$ denotes the entropy of the GMM. Hence, this work studies the problem of maximizing Equation 1 with respect to the parameters $\boldsymbol{\theta}$, which correspond to the weights $q(o)$ and the mean and covariance matrix for each Gaussian component $q(\mathbf{x}|o)$.

Gaussian mixture models are a simple yet powerful choice for a model family since they can approximate arbitrary distributions when assuming a sufficiently high number of components. Compared to more complex models, such as normalizing flows (Kobyzev et al., 2020), they are more interpretable and tractable since not only sampling and evaluating GMMs is cheap, but also marginalizations and certain expectations (of linear or quadratic functions) can be computed in closed form. Furthermore, the simplicity of GMMs allows for sample-efficient learning algorithms, that is, algorithms that require relatively few evaluations of the target distribution for learning the model parameters $\boldsymbol{\theta}$. GMM-based variational inference is, in particular, relevant for problem settings with up to a few hundred dimensions (Volpp et al., 2023; Ewerton et al., 2020; Pignat et al., 2020), where learning and storing of full covariance matrices is still tractable.

Arguably the two most effective algorithms for maximizing Equation 1, both apply independent natural gradient (NG) updates on each component as well as on the categorical distribution over weights (Arenz et al., 2018; Lin et al., 2019a). Yet, both algorithms were derived from a different perspective, have different theoretical guarantees, and even different objectives for the independent updates. Namely, iBayes-GMM (Lin et al., 2019a; 2020) uses the original GMM objective (Eq. 1) for each independent update to perform natural gradient descent also with respect to the full mixture model, whereas VIPS (Arenz et al., 2018; 2020) uses a lower bound for an expectation-maximization procedure, which yields independent objective functions for each component and the mixture weights. Their approach can be shown to converge, even when the M-Step does not consist of single natural gradient updates. However, it was not yet proven, that their proposed procedure, which does use single NG steps, also performs natural gradient descent on the full mixture.

In this work, we will further explore the previous works iBayes-GMM (Lin et al., 2019a; 2020) and VIPS (Arenz et al., 2018; 2020), and use our findings to derive a generalized method that outperforms both of them. In particular, we will present the following contributions.

- In Section 2, we will review and compare the derived update equations of IBayesGMM (Section 2.1) and VIPS (Section 2.2) without considerations of implementation details and design choices. We will prove in Section 2.3 that these update equations are actually equivalent and, therefore, the methods only differ due to design choices not mandated by the derivations. By connecting these two previously separated lines of research, we improve our theoretical understanding of GMM-based VI.

- In Section 3 we will present a general framework for learning a GMM approximation based on the update equations discussed in Section 2. Our framework uses seven modules to independently select design choices, for example, regarding how samples are selected, how natural gradients are estimated or how the learning rate or the number of components is adapted. For each design choice, we will review and compare the different options that have been used in prior works, and we will discuss potential limitations. For example, VIPS uses an inefficient zero-order method for estimating natural gradients, whereas IBayesGMM updates the individual components based on samples from current GMM approximation, which can prevent component with low weight from receiving meaningful updates.

- In Section 4, we evaluate several design choices based on a highly modular implementation of our generalized framework. We propose a novel combination of design choices and show that it significantly outperforms both prior methods. In particular, we combine KL-constrained trust regions, which have been popularized in the gradient-free reinforcement learning setting (Peters et al., 2010; Schulman et al., 2015; Otto et al., 2021), with gradient-based estimates of the NG (Lin et al., 2019b), use samples from each component and adapt the number of components. Test problems are used from both prior works.

- We release the open-source implementation of our generalized framework for GMM-based VI. Our implementation allows each design choice to be set independently and outperforms the reference implementations of iBayes-GMM and VIPS when using their respective design choices. A separate reproducibility package contains the scripts we used for starting each experiment, including hyperparameter optimization.

## 2 Two Derivations for the Same Updates

Maximizing the ELBO in Eq. 1 with respect to the whole mixture model can be reduced to individual updates of the components and the weight distribution (Arenz et al., 2018; Lin et al., 2019a). VIPS (Arenz et al., 2018) achieves this reduction using a lower-bound objective for an expectation-maximization procedure. In contrast, Lin et al. (2019a) investigated the natural gradient of Eq. 1 with respect to the GMM parameters and showed that it can be estimated independently for the parameters of the individual components and the parameters of the weight distribution. While both decompositions can be applied to larger classes of latent variable models, in the following we restrict our discussion to Gaussian mixture models. We will briefly review the different derivations in Section 2.1 and 2.2 before presenting our main theoretical result in Section 2.3, namely, that the derived updates are equivalent. Throughout this section, we deliberately avoid the discussion of design choices and implementation details that are not mandated by the derivations. Instead we will discuss the different options—which may significantly affect the performance—in Section 3.

### 2.1 iBayesGMM: Independently Computing the Natural Gradient

The ELBO objective (Eq. 1) could be straightforwardly optimized using (vanilla) gradient descent, using the reparameterization trick (Kingma and Welling, 2014; Rezende et al., 2014) for obtaining the gradient with respect to the parameters of each component and the weights. However, compared to gradient descent, natural gradient descent (Amari, 1998) has been shown to be much more efficient for variational inference (Khan and Nielsen, 2018). Whereas gradient descent performs steepest descent subject to the constraint of (infinitesimal) small (in the Euclidean sense) changes to the parameters, natural gradient descent performs steepest descent subject to small changes to the underlying distribution (with respect to the Fisher information metric). The natural gradient can be obtained from the vanilla gradient by preconditioning it with the inverse Fisher information matrix (FIM), but explicitly computing the FIM is expensive. Instead, Khan and Nielsen (2018) have shown that the natural gradient with respect to the natural parameters of an exponential family distribution (such as a Gaussian) is given by the vanilla gradient with respect to the

expectation parameters (please refer to Appendix D for additional background). However, GMMs do not belong to the exponential family and are, thus, not directly amenable to this procedure.

To derive efficient natural gradient updates for a broader class of models, Lin et al. (2019a) considered latent variable models, such as GMMs, where the marginal distribution of the latent variable $q(o)$ and the conditional distribution of the observed variable $q(x|o)$ are both minimal exponential family distributions. They showed that for such minimal conditionally exponential family (MCEF) distributions, the Fisher information matrix of the joint distribution $q(x, o)$ is block-diagonal, which in turn justifies computing the natural gradients of the individual components and the weight distribution independently.

## 2.2 VIPS: Independently Maximizing a Lower Bound

Whereas Lin et al. (2019a) showed that a single natural gradient update can be performed for every component independently and that such procedure also performs natural gradient descent on the whole mixture model, Arenz et al. (2018) proposed a method for GMM-based variational inference that is not derived for natural gradient descent and that allows for independent optimizations (going beyond single step updates).

For understanding how VIPS (Arenz et al., 2018) decomposed the ELBO objective (Eq. 1) into independent objectives, it is helpful to first understand which terms prevent independent optimization in the first place. In particular, note that the first term of the ELBO given in Equation 1 already decomposes into independent objectives for each component, and only the second term (the model entropy) prevents independent optimization. More specifically, when using Bayes' theorem to write the probability density of the GMM in terms of the marginals and conditionals, the interdependence between the different components can be narrowed down to the (log-)responsibilities $q_{\boldsymbol{\theta}}(o|\mathbf{x})$ within the entropy,

$$H(q_{\boldsymbol{\theta}}) = -\sum_o q_{\boldsymbol{\theta}}(o) \int_{\mathbf{x}} q_{\boldsymbol{\theta}}(\mathbf{x}|o)\Big( \log \frac{q_{\boldsymbol{\theta}}(o)q_{\boldsymbol{\theta}}(\mathbf{x}|o)}{q_{\boldsymbol{\theta}}(o|\mathbf{x})}\Big)d\mathbf{x},$$

which is the only term in Equation 1 that creates a mutual dependence between different components. Hence, Arenz et al. (2018) introduced an auxiliary distribution $\tilde{q}(o|\mathbf{x})$, to derive the lower bound

$$\tilde{J}(\tilde{q}, \boldsymbol{\theta}) = \sum_o q_{\boldsymbol{\theta}}(o)\Big[ \int_{\mathbf{x}} q_{\boldsymbol{\theta}}(\mathbf{x}|o)\Big( \log \tilde{p}(\mathbf{x}) + \log \tilde{q}(o|\mathbf{x})\Big)d\mathbf{x} + H(q_{\boldsymbol{\theta}}(\mathbf{x}|o))\Big] + H(q_{\boldsymbol{\theta}}(o)) \tag{2}$$

$$= J(\boldsymbol{\theta}) - \int_{\mathbf{x}} q_{\boldsymbol{\theta}}(\mathbf{x})\mathrm{KL}(q_{\boldsymbol{\theta}}(o|\mathbf{x})||\tilde{q}(o|\mathbf{x}))d\mathbf{x}.$$

As an expected KL is non-negative for any two distributions and equals zero, if both distributions are equal, $\tilde{J}(\tilde{q}, \boldsymbol{\theta})$ is a lower bound of $J(\boldsymbol{\theta})$ that is tight when $\tilde{q}(o|\mathbf{x}) = q_{\boldsymbol{\theta}}(o|\mathbf{x})$. Hence, VIPS uses a procedure similar to expectation maximization: In the E-step of iteration $i$, the auxiliary distribution is set to the current model, that is, $\tilde{q}(o|\mathbf{x}) := q_{\boldsymbol{\theta}^{(i)}}(o|\mathbf{x})$. Then, in the M-step, the model parameters are optimized with respect to the corresponding lower bound $\tilde{J}(q_{\boldsymbol{\theta}^{(i)}}, \boldsymbol{\theta})$. As the lower bound is tight before each M-step, i.e, $\tilde{J}(q_{\boldsymbol{\theta}^{(i)}}, \boldsymbol{\theta}^{(i)}) = J(\boldsymbol{\theta}^{(i)})$, it also improves the original objective. As the auxiliary distribution $\tilde{q}(o|\mathbf{x})$ is independent of the model parameters (during the M-step), the weights $q_{\boldsymbol{\theta}}(o)$ and each component $q_{\boldsymbol{\theta}}(\mathbf{x}|o)$ can be optimized independently, resulting in a component-wise loss function

$$\tilde{J}_o(\tilde{q}, \boldsymbol{\theta}) = \int_{\mathbf{x}} q_{\boldsymbol{\theta}}(\mathbf{x}|o)\Big( \log \tilde{p}(\mathbf{x}) + \log \tilde{q}(o|\mathbf{x}) - \log q_{\boldsymbol{\theta}}(\mathbf{x}|o)\Big)d\mathbf{x}. \tag{3}$$

While these derivations would justify any procedure that improves the model with respect to the lower bound objective during the M-step, VIPS, performs single natural gradient steps, closely connecting it to iBayes-GMM. Indeed, we will now show that the updates of both methods are equivalent, except for design choices that are not mandated by the derivations and can be interchanged arbitrarily.

## 2.3 The Equivalence of Both Updates

Both previously described methods iteratively improve the GMM by applying at each iteration a single NG step independently to each component and the weight distribution. Although the NG is computed with

respect to different objective functions, namely the original ELBO (Eq. 1) and the lower bound (Eq. 2), we clarify in Theorem 2.1 that both natural gradients, and, thus, the derived updates, are indeed the same.

**Theorem 2.1.** *Directly after the E-step (which sets the auxiliary distribution to the current model with parameters $\boldsymbol{\theta}^{(i)}$, that is $\tilde{q}(o|\mathbf{x}) \coloneqq q_{\boldsymbol{\theta}^{(i)}}(o|\mathbf{x})$), the natural gradient (denoted by $\tilde{\nabla}$) of the lower bound objective (Eq. 2) matches the natural gradient of the ELBO (Eq. 1), $\tilde{\nabla}_{\boldsymbol{\theta}} J(\boldsymbol{\theta})\big|_{\boldsymbol{\theta}=\boldsymbol{\theta}^{(i)}} = \tilde{\nabla}_{\boldsymbol{\theta}} \tilde{J}(q_{\boldsymbol{\theta}^{(i)}}, \boldsymbol{\theta})\big|_{\boldsymbol{\theta}=\boldsymbol{\theta}^{(i)}}$.*

*Proof.* See Appendix E. $\qquad\qquad\qquad\qquad\qquad\qquad\qquad\qquad\qquad\qquad\qquad\qquad\qquad\square$

This deep connection between both methods was previously not understood and has several implications. First of all, it shows that VIPS actually performs natural gradient descent on the whole mixture model (not only on the independent components) and with respect to the original ELBO objective (not only to the lower bound). On the other hand, it may be interesting to exploit the milder convergence guarantees of VIPS (even bad NG estimates may lead to convergences as long as we improve on the lower bound objective) to perform more robust natural gradient descent. Increasing the robustness by monitoring the lower bound may in particular become relevant when it is difficult to approximate the natural gradient sufficiently well, for example in amortized variational inference, where we want to learn a neural network that predicts the mean and covariances based on a given input. Finally, our new insights raise the question, why the reference implementations of the different methods perform quite differently, despite using the same underlying update equations. In the remainder of this work, we will shed some light on the latter question, by analyzing the different design choices and evaluating their effects on the quality of the learned approximations.

## 3 A Modular and General Framework

We have shown that both derivations suggest exactly the same updates (although the decision to perform single natural gradient steps is voluntary for the EM-based derivation). However, for realizing these updates, several design choices can be made, for example, how to estimate the natural gradient when updating a Gaussian component. By closely comparing both previous methods and their respective implementations, we identified seven design choices that distinguish them. Hence, we can unify both methods in a common framework (Algorithm 1) with seven modules that can be implemented depending on the design choice: **(1)** The *SampleSelector* selects the samples that are used during the current iteration for estimating the natural gradients with respect to the parameters of the components and the weight distribution, **(2)** the *ComponentStepsizeAdaptation* module chooses the stepsizes for the next component updates, **(3)** the *NgEstimator* estimates the natural gradient for the component update based on the selected samples, **(4)** the *NgBasedUpdater* performs the component updates based on the estimated natural gradients and the stepsizes, **(5)** the *WeightStepsizeAdaptation* module chooses the stepsize for the next update of the weights, **(6)** the *WeightUpdater* updates the weight distribution along the natural gradient based on the chosen stepsize, and **(7)** the *ComponentAdaptation* module decides whether to add or delete components. In the remainder of this section, we will discuss and compare the different options that have been employed by prior work.

---

**Algorithm 1** Natural Gradient GMM Variational Inference

> **repeat**
>> samples, targetDensitiesAndGradients ← SampleSelector.selectSamples(GMM, SampleDatabase)
>> compStepsizes ← ComponentStepsizeAdapation.updateStepsizes(numCompUpdates, compStepsizes)
>> naturalGradients ← NGEstimator.getNGs(samples, targetDensitiesAndGradients, GMM)
>> GMM ← NGBasedUpdater.applyNgUpdate(naturalGradients, stepsizes, GMM)
>> weightStepsize ← WeightStepsizeAdaptation.updateStepsizes(numIterations, weightStepsize)
>> GMM ← WeightUpdater.updateWeights(weightStepsize, samples, targetDensities, GMM)
>> GMM ← ComponentAdaptation.adaptNumberOfComponents(numIterations, GMM, SampleDatabase)
> **until** stoppingCriteria

---

### 3.1 Sample Selection

For estimating the natural gradients of the components and the weight distribution, the target distribution $\tilde{p}(\mathbf{x})$ needs to be evaluated on samples from the respective distribution. However, both iBayes-GMM and VIPS use importance sampling to use the same samples for updating all distributions at a given iteration.

**iBayesGMM** obtains the samples for a given iteration by sampling from the current GMM, using standard importance sampling to share samples between different components.

**VIPS** samples from the individual components, rather than the GMM and also stores previous samples (and their function evaluations) in a database. Using two additional hyperparameters, $n_{\text{reused}}$ and $n_{\text{des}}$, VIPS starts by obtaining the $n_{\text{reused}}$ newest samples from the database. Then, the effective sample size $n_{\text{eff}}(o) = \left(\sum_i w_i^{\text{sn}}(o)^2\right)^{-1}$ is computed using the self-normalized importance weights $w_i^{\text{sn}}(o)$, and $n_{\text{des}} - n_{\text{eff}}(o)$ new samples are drawn from each component.

> *In our implementation both options can make use of samples from previous iterations by setting the hyperparameter $n_{reused}$ larger than zero (Arenz et al., 2020). The first option, which generalizes the procedure used by iBayes-GMM (Lin et al., 2020) computes the effective sample size $n_{eff}$ on the GMM and draws $\max(0, N \cdot n_{des} - n_{eff})$ new samples from the GMM. The second option computes $n_{eff}(o)$ for each component and draws $\max(0, n_{des} - n_{eff}(o))$ new samples from each component, matching the procedure used by VIPS.*

### 3.2 Natural Gradient Estimation

For estimating the natural gradient for the individual component updates, VIPS only uses black-box evaluations of the unnormalized target distribution $\log \tilde{p}(\mathbf{x}_i)$ on samples $\mathbf{x}_i$. In contrast, the NG estimate used by iBayes-GMM (Lin et al., 2020), which is based on Stein's Lemma (Stein, 1981), uses first-order information $\nabla_{\mathbf{x}_i} \log \tilde{p}(\mathbf{x}_i)$ which is less general but typically more sample efficient.

**VIPS** uses the policy search method MORE (Abdolmaleki et al., 2015), which is based on compatible function approximation (Pajarinen et al., 2019; Peters and Schaal, 2008; Sutton et al., 1999). Namely, as shown by Peters and Schaal (2008), an unbiased estimate of the natural gradient for an objective of the form $\mathbb{E}_{\pi_{\boldsymbol{\theta}}}(\mathbf{x})[R(\mathbf{x})]$ is given by the weights $\boldsymbol{\omega}$ of a compatible function approximator (Sutton et al., 1999), $\tilde{R}(\mathbf{x}) = \boldsymbol{\omega}^\top \boldsymbol{\phi}(\mathbf{x})$, that is fitted using ordinary least squares to approximate $R$, based on a data set $\mathcal{X} = \{(\mathbf{x}, R(\mathbf{x}))_i\}$, with samples $\mathbf{x}_i$ that are obtained by sampling from the distribution $\pi_{\boldsymbol{\theta}}$. A function approximator is compatible to the distribution $\pi_{\boldsymbol{\theta}}$, if the basis functions $\boldsymbol{\phi}(\mathbf{x})$ are given by the gradient of the log distribution, $\boldsymbol{\phi}(\mathbf{x}) = \nabla_{\boldsymbol{\theta}} \log \pi_{\boldsymbol{\theta}}(\mathbf{x})$. For updating a Gaussian component $q(\mathbf{x}|o)$ parameterized with its natural parameters, $\boldsymbol{\eta}_o = \left\{\boldsymbol{\Sigma}_o^{-1}\boldsymbol{\mu}_o, -\frac{1}{2}\boldsymbol{\Sigma}_o^{-1}\right\}$, the compatible function approximator can be written as

$$\tilde{R}(\mathbf{x}) = \mathbf{x}^\top \mathbf{R} \mathbf{x} + \mathbf{x}^\top \mathbf{r} + r,$$

where the matrix $\mathbf{R}$, the vector $\mathbf{r}$ and the scalar $r$, are the linear parameters that are learned using least squares. Here, the constant offset $r$ can be discarded and $\mathbf{R}$ and $\mathbf{r}$ directly correspond to the natural gradients, that could be used to update the natural parameters,

$$-\frac{1}{2}\boldsymbol{\Sigma}_o^{-1} = -\frac{1}{2}\boldsymbol{\Sigma}_{o,\text{old}}^{-1} + \beta_o \mathbf{R}, \qquad \boldsymbol{\Sigma}_o^{-1}\boldsymbol{\mu}_o = \boldsymbol{\Sigma}_{o,\text{old}}^{-1}\boldsymbol{\mu}_{o,\text{old}} + \beta_o \mathbf{r}. \qquad (4)$$

The least squares targets based on the ELBO (Eq. 1), $R_{\text{ELBO}}(\mathbf{x}) = \log \tilde{p}(\mathbf{x}) - \log q_{\boldsymbol{\theta}}(\mathbf{x})$, and the least squares targets based on the lower bound (Eq. 2), $R_{\text{LB}}(\mathbf{x}) = \log \tilde{p}(\mathbf{x}) + \log \tilde{q}(o|\mathbf{x}) - \log \tilde{q}(\mathbf{x}|o)$ only differ by a constant offset $\log q_{\boldsymbol{\theta}}(o)$ that would be consumed by the scalar $r$ and not affect the NG estimates $\mathbf{R}$ and $\mathbf{r}$.

For compatible function approximation, the data set $\mathcal{X}$ should be obtained by sampling the respective component. However, VIPS uses importance weighting to make use of samples from previous iterations and from different components, by performing weighted least squares, weighting each data point by its importance weight. The importance weights $w_i$ can be computed as $\frac{q(\mathbf{x}_i|o)}{z(\mathbf{x}_i)}$, where $z$ is the actual distribution that was used for sampling the data set. However, VIPS uses self-normalized importance weights $w_i^{\text{sn}} = (\sum_j w_j)^{-1} w_i$, which yield lower-variance, biased, but asymptotically unbiased estimates. Furthermore, VIPS applies ridge regularization during weighted least squares, which further introduces a bias.

**iBayesGMM** exploits that for exponential-family distributions, the natural gradient with respect to the natural parameters $\boldsymbol{\eta}$ (for Gaussians $\boldsymbol{\eta} = \{\boldsymbol{\Sigma^{-1}\mu}, -\frac{1}{2}\boldsymbol{\Sigma^{-1}}\}$) corresponds to the vanilla gradient with respect to the expectation parameters $\mathbf{m}$ (for Gaussians $\mathbf{m} = \{\boldsymbol{\mu}, \boldsymbol{\Sigma} + \boldsymbol{\mu}\boldsymbol{\mu}^\top\}$) (Khan and Nielsen, 2018). Using the chain rule, the gradient with respect to the expectation parameters can be expressed in terms of the gradient with respect to the covariance $\boldsymbol{\Sigma}$ and mean $\boldsymbol{\mu}$ (Khan and Lin, 2017, Appendix B.1). Thus, the NG step for a Gaussian component $q(\mathbf{x}|o)$ with stepsize $\beta_o$ and objective $J(\mathbf{x})$ can be computed as

$$-\frac{1}{2}\boldsymbol{\Sigma}_o^{-1} = -\frac{1}{2}\boldsymbol{\Sigma}_{\text{o,old}}^{-1} + \beta_o \nabla_{\boldsymbol{\Sigma}_o} J, \quad \boldsymbol{\Sigma}_o^{-1}\boldsymbol{\mu}_o = \boldsymbol{\Sigma}_{\text{o,old}}^{-1}\boldsymbol{\mu}_{\text{o,old}} + \beta_o \left(-2\Big[\nabla_{\boldsymbol{\Sigma}} J\Big]\boldsymbol{\mu}_{\text{o,old}} + \nabla_{\boldsymbol{\mu}} J\right), \tag{5}$$

as shown by Khan et al. (2018, Appendix C). As the objective $J$ corresponds to an expected value, i.e., $J = \mathbb{E}_{q(\mathbf{x}|o)}\Big[R(\mathbf{x})\Big]$ with $R(\mathbf{x}) = \log\frac{\tilde{p}(\mathbf{x})}{\tilde{q}(\mathbf{x})}$, the gradient with respect to mean and covariance are given by the expected gradient and Hessian (Opper and Archambeau, 2009),

$$\nabla_{\boldsymbol{\Sigma}} J = \frac{1}{2}\mathbb{E}_{q(\mathbf{x}|o)}\Big[\nabla_{\mathbf{xx}} R(\mathbf{x})\Big], \qquad \nabla_{\boldsymbol{\mu}} J = \mathbb{E}_{q(\mathbf{x}|o)}\Big[\nabla_{\mathbf{x}} R(\mathbf{x})\Big]. \tag{6}$$

Hence, using Monte-Carlo to estimate the gradients with respect to mean and covariance (Eq. 6), we can obtain unbiased estimates of the natural gradient (Eq. 5). However, evaluating the Hessian $\nabla_{\mathbf{xx}} R(\mathbf{x})$ is computationally expensive, and therefore, Lin et al. (2019b) proposed an unbiased first-order estimate of the expected Hessian based on Stein's Lemma (Stein, 1981) given by (Lin et al., 2019b, Lemma 11)

$$\mathbb{E}_{q(\mathbf{x}|o)}\Big[\nabla_{\mathbf{xx}} R(\mathbf{x})\Big] = \mathbb{E}_{q(\mathbf{x}|o)}\big[\boldsymbol{\Sigma}_o^{-1}(\mathbf{x} - \boldsymbol{\mu}_o)\left(\nabla_{\mathbf{x}} R(\mathbf{x})\right)^\top\big]. \tag{7}$$

Similar to VIPS, iBayes-GMM uses importance sampling to perform the Monte-Carlo estimates (Eq. 6 right, Eq. 7) based on samples from different components. However, in contrast to VIPS, iBayes-GMM uses standard (unbiased) importance weighting, rather than self-normalized importance weighting, which in our implementation, can be selected using a hyperparameter that is available for both options.

> *Our implementation supports both options, where using standard importance weighting or self-normalized importance weighting can be chosen using a hyperparameter that is available to both options.*

### 3.3 Natural Gradient based Component Updates

For performing the natural gradient update, we identified three different options in the related literature. Lin et al. (2019a) directly apply the natural gradient update (Eq. 4) based on the natural gradients $\mathbf{R}$, $\mathbf{r}$ and the stepsize $\beta_o$, which we assume given. However, this update may lead to indefinite covariance matrices, and therefore iBayes-GMM (Lin et al., 2020) uses the *improved Bayesian learning rule (iBLR)*, which modifies the update direction (no longer performing natural gradient descent) in a way that ensures that the covariance matrix remains positive definite. Thirdly, VIPS indirectly control the stepsize by choosing an upper bound on the KL divergence between the updated and the old component, $\text{KL}(q(\mathbf{x}|o)||q_{\text{old}}(\mathbf{x}|o))$, and solves a convex optimization problem to find the largest $\beta_o$ that satisfies the trust region constraint. Directly performing the natural gradient step is straightforward, and therefore, we now only discuss the two latter options.

**iBayesGMM** applies the improved Bayesian Learning Rule (iBLR), which approximates Riemannian gradient descent (Lin et al., 2020),

$$-\frac{1}{2}\boldsymbol{\Sigma}_o^{-1} = -\frac{1}{2}\boldsymbol{\Sigma}_{\text{o,old}}^{-1} + \beta_o(\mathbf{R} - \beta_o\mathbf{R}\boldsymbol{\Sigma}_{\text{o,old}}\mathbf{R}), \qquad \boldsymbol{\Sigma}_o^{-1}\boldsymbol{\mu}_o = \boldsymbol{\Sigma}_o^{-1}\boldsymbol{\Sigma}_{\text{o,old}}\left(\boldsymbol{\Sigma}_{\text{o,old}}^{-1}\boldsymbol{\mu}_{\text{o,old}} + \beta_o\mathbf{r}\right). \tag{8}$$

The iBLR update (Eq. 8) differs from the NG update (Eq. 4) due to the additional terms $-\beta_o\mathbf{R}\boldsymbol{\Sigma}_{\text{o,old}}\mathbf{R}$ and $\boldsymbol{\Sigma}_o^{-1}\boldsymbol{\Sigma}_{\text{o,old}}$. Using this update, the new precision matrix can be shown to be an average of a positive definite and a positive semidefinite term, $\boldsymbol{\Sigma}_o^{-1} = \frac{1}{2}\left(\boldsymbol{\Sigma}_{\text{o,old}}^{-1} + \mathbf{U}^\top\mathbf{U}\right)$, where $\mathbf{U} = \mathbf{L} - 2\beta\mathbf{L}^{-\top}\mathbf{R}$ and $\mathbf{L}^\top\mathbf{L} = \boldsymbol{\Sigma}_{\text{o,old}}^{-1}$.

**VIPS** updates the components using natural gradient descent (Eq. 4). However, rather than controlling the change of the component using the stepsize $\beta_o$, it upper-bounds the KL divergence $\text{KL}\left(q_{\boldsymbol{\theta}}(\mathbf{x}|o)||\tilde{q}(\mathbf{x}|o)\right) < \epsilon_o$ by solving for every component update an optimization problem that finds the largest stepsize $\beta_o$, such that

the updated component will have a positive definite covariance matrix and a KL divergence to the original component that is smaller than a bound $\epsilon_o$, which is assumed given instead of the stepsize $\beta_o$ (Abdolmaleki et al., 2015). The additional optimization problems for finding the stepsize add little computational overhead since the KL divergence is convex in the stepsize and can be efficiently computed for Gaussian distributions.

> *Our implementation supports three options: (1) directly applying the NG update (Eq. 4), (2) the iBLR update, and the (3) trust-region update of VIPS. However, VIPS uses an external L-BFGS-G optimizer to find the largest stepsize that respects the trust-region, which was difficult to efficiently integrate into our Tensorflow (Abadi et al., 2015) implementation. Instead we implemented a simple bisection method so solve this convex optimization problem, see Appendix I for details.*

## 3.4 Weight Update

The natural gradient update for the categorical distribution is performed by updating the log weights in the direction of the expected reward of the corresponding component, that is,

$$q(o) \propto q(o)_{\text{old}} \exp\left(\beta \hat{R}(o)\right), \tag{9}$$

where $\hat{R}(o)$ is a Monte-Carlo estimate of $R(o) = \mathbb{E}_{q(\mathbf{x}|o)}\left[\log \frac{p(\mathbf{x})}{\tilde{q}(\mathbf{x})}\right]$. The NG update is presented differently by Lin et al. (2020), but we clarify in Appendix F that both updates are indeed equivalent[1]. However, we identified two different options to perform the weight update.

**IBayesGMM** directly applies the stepsize $\beta$.

**VIPS** originally optimized the stepsize with respect to a desired KL bound KL $(q(o)||q_{\text{old}}(o)) \leq \epsilon$, analogously to the component update described in Section 3.3 (Arenz et al., 2018). Although in subsequent work, Arenz et al. (2020) found that a fixed stepsize $\beta = 1$, corresponding to a greedy step to the optimum of their lower bound objective, performs similarly well.

> *Our implementation supports both options, directly applying the stepsize and optimizing the stepsize with respect to a trust-region. Directly applying the stepsize in combination with a WeightStepsizeAdaption that always selects a fixed stepsize $\beta = 1$ corresponds to the greedy procedure (Arenz et al., 2020).*

## 3.5 Weight and Component Stepsize Adaptation

While we use separate modules for *WeightStepsizeAdaptation* and *ComponentStepsizeAdaptation*, we consider the same adaptation schemes in both modules.

**iBayesGMM** used fixed stepsizes in their implementation but used different stepsizes for weights and for the components. Often, a weight stepsize $\beta = 0$ was used to maintain a uniform distribution over the different components. We believe that this concession was probably required due to their design choice, of using samples from the GMM for updating the individual components, which may prevent components with low weight to obtain meaningful updates. In prior work, Khan and Lin (2017) discussed a backtracking method to adapt the stepsize for the component updates for ensuring positive definiteness, but they found it slow in practice and did not test it in the GMM setting, and Khan et al. (2018) used a decaying stepsize.

**VIPS** originally used fixed trust-regions for the weight and component update (Arenz et al., 2018). Later a fixed stepsize with $\beta = 1$ for the weights was combined with adaptive trust-regions for the component updates, where the bounds were chosen independently for each component, by increasing the bound if the reward of the corresponding component improved during the last update, and by decreasing it otherwise.

> *Our implementation supports three options for each stepsize scheduler: (1) keeping the stepsize constant (2) using a decaying stepsize (Khan et al., 2018) and (3) adapting the stepsize based on the improvement of the last update (Arenz et al., 2020).*

---

[1]The respective implementations still differ slightly because VIPS uses self-normalized importance weighting, rather than standard importance-weighting, which we control in our implementation using a hyperparameter.

| MODULE | OPTIONS | | | | | |
|---|---|---|---|---|---|---|
| NgEstimator | More | **Z** | Stein | **S** | | |
| ComponentAdaptation | Fixed | **E** | Vips | **A** | | |
| SampleSelector | Lin et al. | **P** | Vips | **M** | | |
| NgBasedComponentUpdater | Direct | **I** | iBLR | **Y** | Trust-Region | **T** |
| ComponentStepsizeAdaptation | Fixed | **F** | Decaying | **D** | Adaptive | **R** |
| WeightUpdater | Direct | **U** | Trust-Region | **O** | | |
| WeightStepsizeAdaptation | Fixed | **X** | Decaying | **G** | Adaptive | **N** |

Table 1: We assign a unique letter to every option such that every combination of options can be specified with a 7-letter word (one letter per module).

## 3.6 Component Adaptation

**VIPS** dynamically adapts the number of components by deleting components at poor local optima that do not contribute to the approximation, and by adding new components in promising regions. The initial mean for the new component is chosen based on a single-sample estimate of the initial reward $\tilde{R}(o_{\text{new}})$ for the new component, evaluated for samples from the database.

**iBayes-GMM** uses a fixed number of components $K$ that is specified by a hyperparameter.

> *Our implementation supports (1) a dummy option that does not do anything and (2) adapting the number of components dynamically (Arenz et al., 2020).*

## 4 Experiments

To evaluate the effect of the individual design choices, we implemented all previously discussed options in a common framework, such that they can be chosen independently. Prior to the experiments we ensured that our implementation performs comparably to the MATLAB implementation of iBayes-GMM (Lin et al., 2020) and to the C++-implementation of VIPS (Arenz et al., 2020) when using the respective design choices. As shown in Appendix G, when comparing our implementation with the reference implementation on their target distributions, we always learn similar or better approximations. Additional details on the implementation are described in Appendix I. An overview of the available options is shown in Table 1. Available hyperparameters for each option are shown in Appendix H. Our framework allows for $2^4 \cdot 3^3 = 432$ different combinations of options. We assign a unique letter to each option (see Table 1), such that we can refer to each of the combinations with a 7-letter codeword. For example, SEPYFUX refers to iBayes-GMM (Lin et al., 2020) and ZAMTRUX to VIPS (Arenz et al., 2020). However, evaluating each combination is computationally prohibitive. Instead, we designed the following sequence of experiments.

In the first group of experiments, we evaluate the stability of the component updates, using fixed weights, and without adapting the number of components. In the second group of experiments, we use the most promising design choices for the component update that we identified in the first group of experiments, and ablate over the design choices that affect the update of the weights and the adaptation of the number of components. Based on the results of the previous group of experiments, we then identify promising combinations and evaluate them on the full test suite. For comparison with prior work, we also evaluate the design choices used by iBayes-GMM (Lin et al., 2020) and VIPS (Arenz et al., 2020).

### 4.1 Experiment 1: Component Update Stability

For evaluating the effects of the design choices on the component update stability, we do not change the number of components (Option **E**) and do not update the (uniformly initialized) weights (**X** and **U**, with initial weight-stepsize $\beta = 0$). Furthermore, we always sample from the GMM (**P**), since the two options for the *SampleSelector* hardly differ for uniform GMMs. We evaluate the different options for the *NgEstimator*, *ComponentStepsizeAdaptation* and *NgBasedComponentUpdater* resulting in 18 different combinations.

| Design Choice | | BreastCancer | BreastCancerMB | Wine |
|---|---|---|---|---|
| Component NG Estimation | MORE (Z) | 78.88 | 152.84 | N/A |
| | Stein (S) | **78.46** | **68.22** | **1431.09** |
| Component Update | Direct (I) | 78.95 | 70.96 | 2263.78 |
| | iBLR (Y) | 78.69 | 70.37 | **1431.09** |
| | Trust-Region (T) | **78.46** | **68.22** | 1443.40 |
| Component Stepsize Adaptation | Fixed (F) | **78.46** | 69.01 | 1450.11 |
| | Decaying (D) | 78.47 | 70.69 | 1468.50 |
| | Adaptive (R) | **78.46** | **68.22** | **1431.09** |

Table 2: We show optimistic estimates of the best performance (negated ELBO) that we can achieve with every option when optimizing a uniform GMM with a fixed number of components, by reporting the best performance achieved during Bayesian optimization among all runs that used the respective option.

We evaluate each candidate on three target distributions: In the *BreastCancer* experiment Arenz et al. (2018) we perform Bayesian logistic regression using the full "Breast Cancer" datatset (Lichman, 2013). Further, we introduce the variant *BreastCancerMB* which uses minibatches of size 64 to evaluate the effect of stochastic target distributions. Thirdly, we evaluate the candidates in a more challenging Bayesian neural network experiment on the "Wine" dataset (Lichman, 2013), where we use a batch size of 128. We use two hidden layer with width 8, such that the 177-dimensional posterior distribution is still amenable to GMMs with full covariance matrices. Please refer to Appendix J for details on the implementation for all experiments.

We tuned the hyperparameters of each of the 18 candidates separately for each test problem using Bayesian optimization. We granted each candidate exclusive access to a compute node with 96 cores for two days per experiment. The results of the hyperparameter search are summarized in Table 2, where we report the best ELBO achieved for every design choice (maximizing over all tested hyperparameters and all combinations of the remaining options). On *Wine*, which uses almost three times as many parameters as have been tested by Arenz et al. (2020), we could not obtain reasonable performance when using MORE, as this zero-order method requires significantly more samples, resulting in very slow optimization.

The values in Table 2 give an optimistic estimate of the best performance that we can expect for a given design choice, however, it is in general not guaranteed that this performance can be robustly achieved over different seeds. Hence, for the most promising candidates, SEPTRUX and SEPYRUX, we used the best hyperparameters from the Bayesian optimization and evaluated the performance over ten seeds. The mean of the final performance and its 99.7% standard error are shown in Table 3. We also show for every experiment a second metric, where we use the maximum mean discrepancy (Gretton et al., 2012) (MMD) for *BreastCancer* and *BreastCancerMB* (comparing samples from the GMM with baseline samples (Arenz et al., 2020)), and the mean squared error of the Bayesian inference predictions for *Wine*. Unfortunately, the hyperparameters for SEPYRUX on *BreastCancer* and *BreastCancerMB* led to unstable optimization for some seeds, despite their good performance during hyperparameter search. We expect that using slightly more conservative hyperparameters, SEPYRUX could reliably achieve a performance that is only slightly worse than the optimistic value provided in Table 3 for *BreastCancer* (78.69). However, the performance of SEPTRUX is even in expectation over multiple seeds already better than this optimistic estimate. Furthermore, also on *WINE*, where SEPYRUX did not suffer from instabilities, the trust-region updates achieved better performance, albeit not statistically significant. Hence, we decided to use trust-region updates for the second group of experiments. We will further test the iBLR update in our main experiments for a more thorough comparison.

## 4.2 Experiment 2: Weight Update and Exploration

According to the first experiment, first-order estimates using Stein's Lemma, adaptive component stepsizes and trust-region updates are the most effective and stable options for the component updates, and therefore, we fix the corresponding design choices ($\mathbf{S}$, $\mathbf{T}$, $\mathbf{R}$) for our second group of experiments. The modules that we did not evaluate in the first group of experiments (*ComponentAdaptation*, *SampleSelector*, *WeightUpdater* and *WeightStepsizeAdaptation*) are in particular relevant for discovering and approximating multiple modes

| Candidate | BreastCancer | | BreastCancerMB | | Wine | |
| | -ELBO | MMD | -ELBO (full batch) | MMD | -ELBO | MSE |
|---|---|---|---|---|---|---|
| SEPYRUX | 1042.28 ±2699.32 | 0.004 ±0.002 | 1130.81 ±799.66 | 0.033 ±0.014 | 1462.91 ±35.70 | 0.481 ±0.022 |
| SEPTRUX | **78.53** **±0.02** | **0.002** **±0.000** | **81.21** **±1.13** | **0.002** **±0.000** | 1444.01 ±30.78 | 0.478 ±0.021 |

Table 3: We evaluated the best hyperparameters for the most promising candidates of our experiments for Group 1 on 10 different seeds with respect to ELBO and a secondary metric (maximum mean discrepancy or mean squared error). SEPYRUX, which uses the iBLR update for the component updates did not achieve stable results on the breast cancer experiments. SEPTRUX which uses trust-region updates for the components outperformed SEPYRUX in all experiments, although in *WINE* the advantage is not statistically significant.

| Design Choice | | GMM20 | STM20 | PlanarRobot4 |
|---|---|---|---|---|
| Component Adaptation | Non-Adaptive (E) | **0.00** | 0.08 | 12.33 |
| | Adaptive (A) | **0.00** | **0.05** | **12.15** |
| Sample Selection | From Mixture (P) | 0.11 | 1.52 | 12.77 |
| | From Components (M) | **0.00** | **0.05** | **12.15** |
| Weight Update | Natural Gradient Descent (U) | **0.00** | 0.06 | 12.30 |
| | Trust-Region (O) | **0.00** | **0.05** | **12.15** |
| Weight Stepsize Adaptation | Fixed (X) | **0.00** | 0.06 | **12.15** |
| | Decaying (G) | **0.00** | **0.05** | 12.33 |
| | Improvement-Based (N) | **0.00** | 0.06 | 12.22 |

Table 4: We show optimistic estimates of the best performance (negated ELBO) that we can achieve with every option (updating the components using the design choices identified in the first experiment). We report the best performance achieved during Bayesian optimization among all runs that used the respective option.

of the target distribution. Hence, we focus on multi-modal target distributions for the second set of experiments. All test problems for these experiments were taken from prior work. Namely, we chose *GMM20* and *PlanarRobot4* from Arenz et al. (2020) and *STM20* from Lin et al. (2020). For *GMM20* and *STM20* the target distribution is given by an unknown mixture of 20-dimensional Gaussians and Student-Ts, respectively. In *PlanarRobot4* we want to approximate a distribution over joint configurations of a 10-link planar robot, such that it approximately reaches any of four possible goal positions. For Bayesian optimization of the hyperparameters, we grant each of the 24 candidates exclusive access to our compute node for one day per test problem. Optimistic estimates of the best performance for each design choices are shown in Table 4. Based on our experiments sampling according to the mixture weights (Option **P**) seems to be clearly inferior to sampling from the components (Option **M**), as it was the only option that was not able to solve the *GMM20* experiment. Furthermore, adapting the number of components (Option **A**) and using trust-region updates for the weight update (Option **O**) seem beneficial for multimodal target distributions. However, for *WeightStepsizeAdaptation*, also a fixed stepsize (when used as trust-region) achieved good performance.

### 4.3 Experiment 3: Evaluating the Promising Candidates

For our main experiment we focus on candidates that use first order NG estimates, adaptive number of components and adaptive component-stepsizes and sample from the individual components (Options **S**, **A**, **M**, **R**), and aim to better compare trust-region updates with the iBLR update (Option **T** vs. **Y**). For updating the weights, we evaluate fixed and adaptive trust-regions, and fixed direct NG updates (**OX** vs. **ON** vs. **UX**), resulting in six candidates: SAMTRUX, SAMTROX, SAMTRON, SAMYRUX, SAMYROX and SAMYRON. Furthermore, to compare with prior work, we also evaluate ZAMTRUX (Arenz et al., 2020) and SEPYFUX (Lin et al., 2020) as well as the variant SEPYRUX, combining iBLR updates with adaptive stepsizes.

| Candidate | BreastCancer | BreatCancerMB | GermanCredit | GermanCreditMB | PlanarRobot | GMM20 | GMM100 | STM20 | STM300 | WINE | TALOS |
|---|---|---|---|---|---|---|---|---|---|---|---|
| Samtron | **78.00** ±**0.02** | **78.41** ±**0.03** | **585.10** ±**0.00** | **585.12** ±**0.00** | **11.47** ±**0.04** | **−0.00** ±**0.00** | **0.01** ±**0.03** | **0.00** ±**0.00** | **14.96** ±**0.48** | **1423.12** ±**31.55** | **−24.43** ±**0.16** |
| Samyron | 78.61 ±0.05 | 79.96 ±0.26 | 585.11 ±0.01 | **585.12** ±**0.00** | 12.93 ±0.13 | **0.00** ±**0.00** | 0.16 ±0.09 | 0.03 ±0.01 | 22.50 ±0.15 | **1433.87** ±**30.49** | −24.00 ±0.23 |
| iBayes-GMM (Sepyfux) | 79.78 ±0.40 | 80.13 ±0.69 | 585.12 ±0.01 | **585.12** ±**0.00** | 17.26 ±2.13 | 0.43 ±0.15 | 4.54 ±0.52 | 0.46 ±0.06 | 26.87 ±0.45 | 3279.32 ±1425.12 | −16.64 ±5.26 |
| VIPS (Zamtrux) | 78.14 ±0.01 | 83.65 ±0.97 | **585.10** ±**0.00** | 585.35 ±0.03 | **11.48** ±**0.04** | **−0.00** ±**0.00** | 1.21 ±0.15 | 0.54 ±0.17 | N/A | 16 503.38 ±813.75 | −23.69 ±0.16 |

Table 5: Along with the negated ELBO, we show the $3\sigma$ confidence intervals based on the standard error of its mean using ten different seeds. The proposed candidate clearly outperforms the prior methods VIPS (Arenz et al., 2020) and ıBayes-GMM (Lin et al., 2020). First-order natural gradient estimates with trust-region constraints (**T**) seem preferable over the iBLR update (**Y**). We observed instabilities for Sepyfux on *PlanarRobot* and *TALOS* and, thus, removed bad outliers when computing the reported values.

We evaluate these candidates on the full test suite, which uses the following test problems on top of the previously discussed ones: *GermanCredit* (Arenz et al., 2020) and *GermanCreditMB* are similar to the *BreastCancer* experiment, but use the 25-dimensional *GermanCredit* dataset (Lichman, 2013); *GMM100* and *STM300* are higher-dimensional variants of the *GMM20* and *STM20* experiments; and, finally, *TALOS* is a new experiment that we introduced to evaluate the different options on a test problem that was not used during development. Somewhat related to the *PlanarRobot4* experiment, in the *TALOS* experiment we aim to learn a distribution over joint configurations to reach a goal position with the robot's endeffector. However, the *TALOS* experiment uses the kinematics of an actual robot–the humanoid TALOS from PAL Robotics (Stasse et al., 2017). Furthermore, the target distribution is based on the work by Pignat et al. (2020) and is more complex by using a product of expert that penalize pose errors for each foot and hand as well as unstable configurations (that occur when the robot's center of mass projected on the ground is outside of the support polygon spanned by the feet), and that incorporate a prior distribution over the joint angles. The target distribution is 34 dimensional (7 joint configurations for each leg, 6 joint angles for each arm and 6 additional parameters that specify the pose of the torso with respect to a fixed reference frame. On *TALOS*, prior to the experiments of this group, we only performed few experiments using SEPIFOX to optimize a single Gaussian component, to ensure that the target distribution is correctly implemented.

For selecting the hyperparameters, we perform for each candidate and each experiment a small grid search, where we make use of the results from the previous experiments to select suitable ranges. Extensive hyperparameter search (e.g. using Bayesian optimization) would unfairly benefit options with more hyperparameters. Table 5 compares the final performance (negated ELBO) of the best-performing candidate (Samtron) with the prior methods VIPS (Arenz et al., 2020) and iBayes-GMM (Lin et al., 2020). We also show Samyron for comparing trust-region natural-gradient updates with the iBLR update. According to these experiments, we can clearly improve upon Zamtrux by using Stein's Lemma for estimating the natural gradient (in particular for higher-dimensional problems), and upon Sepifux, by sampling from the components and adapting their number during optimization. Interestingly, trust-region constraints seem to be beneficial also when using first-order estimates of the natural gradient, and showed a slight but consistent advantage compared to the iBLR update (Lin et al., 2020) in our experiments. Full results of our main experiments are shown in Appendix K, where we show the performance of all tested candidates, also with respect to secondary metrics.

## 5 Conclusion

Although VIPS and iBayes-GMM are derived from different perspectives—where the derivations for Bayes-GMM are less general (by requiring single NG steps for the component update) but enjoy stronger guarantees (by proving natural gradient descent on the whole mixture model)—, we showed that both algorithms only differ in design choices and could have been derived from the other perspective, respectively. This unification of both perspective shows that we can derive approximate natural gradient descent algorithms also for mixtures of non-Gaussian components—where the approximation errors of the natural gradient are potentially much larger—without having to give up on convergence guarantees. Furthermore, our results are of high relevance for the practitioner, both due to our extensive study on the effects of the individual design choices—which shows that both prior works can be improved by using a combination of their design choices—and by releasing our modular framework for natural gradient GMM-based variational inference, which is well-documented and easy to use and outperforms the reference implementations by Arenz et al. (2020) and Lin et al. (2020) when using the respective design choices. Typical trade-offs for the practitioner are discussed in Appendix M. The limitations of this study and the potential for negative societal impact are discussed in Appendix A and B.

## Acknowledgements

This research was supported by "The Adaptive Mind", funded by the Excellence Program of the Hessian Ministry of Higher Education, Science, Research and Art.

The authors gratefully acknowledge the computing time provided to them on the high-performance computer Lichtenberg at the NHR Centers NHR4CES at TU Darmstadt. This is funded by the Federal Ministry of Education and Research, and the state governments participating on the basis of the resolutions of the GWK for national high performance computing at universities.

The authors acknowledge support by the state of Baden-Württemberg through bwHPC.

This work was performed on the HoreKa supercomputer funded by the Ministry of Science, Research and the Arts Baden-Württemberg and by the Federal Ministry of Education and Research.

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

## A    Limitations

The scope of this work is narrow, focusing on two specific approaches for natural-gradient GMM-based variational inference. There are of course many other models that can be applied for variational inference, and, depending on the problem setting, some of these models are highly preferable over GMMs, for example, normalizing flows should likely be preferred for high-dimensional problem settings, such as (deep) Bayesian neural networks. However, for this work we assume that we indeed want to optimize a Gaussian mixture model, for example, because we require an interpretable model with smooth gradients (Ewerton et al., 2020). Even in the field of GMM-based variational inference, alternative methods, based on boosting (Miller et al., 2017; Guo et al., 2016) or the reparameterization trick are possible. By not using natural gradients, these methods can be applied more straightforwardly to sparse covariance parameterizations which can be beneficial for higher-dimensional problems. However, in the considered problem setting where we can learn GMMs with full covariance matrices, these methods are not competitive (Arenz et al., 2020) to the natural gradient based methods described in this work. Please refer to Appendix C for a discussion of related work.

Regarding our empirical study, we want to point out several aspects that could lead to misinterpretations of the results. While we also report a secondary metric for each test problem in our main experiment (Table 8), please recall that the hyperparameter optimization was performed only with respect to the ELBO. We noticed on the *planar robot* experiment, that SAMTRUX could still achieve competitive ELBO when initializing the mixture model with fewer components than we used for our evaluation, while resulting in a significantly worse MMD. Hence, one should consider that for each method, the performance with respect to the secondary metric could potentially be better, if the hyperparameters were chosen correspondingly. Similarly, one should be careful when comparing the learning curves (which can be found in Appendix L), with respect to efficiency or stability, as the hyperparameters have only been chosen based on the final ELBO of each run. We used this simple and hard criterion for selecting the hyperparameters, to make the experimental study more transparent and objective by removing human influence.

Human influence, of course, could not be completely avoided: For our main experiment, we only allowed for a coarse hyperparameter search over carefully chosen parameters. In contrast to the first two groups of experiments, where we used extensive Bayesian optimization to identify the best performance that we can expect for each design choice, in the main experiment we took into account that the computational budget for hyperparameter search (also ours) is limited. As a consequence, the results of our main experiment, are to some extend affected by our ability to propose good hyperparameter ranges for the different design choices. However, we had good prior knowledge about suitable parameter ranges based on the previous groups of experiments and also based on the parameters reported by prior work, and furthermore the results of our main experiment are consistent with the performances we observed in the previous experiments (which used much less subjective Bayesian optimization), and, hence, we conclude that the effect of the human factor is rather small. To increase the transparency of our empirical study we release a separate reproducibility package[2], which contains scripts for running each experiment, thereby documenting the exact conditions under which all our experiments have been started (including hyperparameter search).

## B    Potential for Negative Societal Impact

Machine learning methods can have a significant impact on our daily lives. They can have a positive impact on society by taking work off our hands, addressing challenges like the climate crisis, or helping to develop better medical treatments. But they can also have a negative impact on society by reinforcing prejudices, discriminating against people, invading our privacy, wasting enormous amounts of energy, or exacerbating the imbalance of power and wealth. They can cause serious harm if we overestimate their capabilities, and they can be used maliciously, for example, to falsify data or carry out cyberattacks.

In terms of negative impact, this work is unlikely to contribute significantly to energy waste because we focus on structured representations with few parameters that can be learned efficiently. Furthermore, our work does not focus on processing or forgery of images, text, or speech, so it is unlikely to have a significant negative impact on society due to privacy violations or misinformation. We are confident that we can use

---

[2]https://github.com/OlegArenz/gmmvi_reproducibility

our findings to help people in their daily lives and have a net positive impact on society, although we of course can not foresee how our findings will be used by future work.

## C  Related Work

In addition to the aforementioned methods, several other methods have explored natural gradients and trust regions for variational inference. For mean-field approximations, Hoffman et al. (2013) propose an efficient method for estimating the natural gradient from mini-batches, and Theis and Hoffman (2015) proposed a related KL-constrained trust-region method. Regier et al. (2017) proposed a second-order method for Gaussian variational inference based on Euclidean trust regions. Khan et al. (2015) introduced a KL-based proximal point method for conjugate models and related it to natural gradient descent. For non-conjugate models, they proposed local linearizations. Khan et al. (2016) extended this approach to other divergences and stochastic gradients. Salimbeni et al. (2018) compute the natural gradient based on the Jacobian of the parameters of the Gaussian and its expectation parameters. The Jacobian can be computed using forward-mode differentiation, or by using reverse-mode differentiation twice. For GMM-based variational inference, Daudel et al. (2023) explored the optimization of alternate divergences by presenting a method for minimizing alpha divergences. Alpha divergences are a family of divergences parameterized by a scalar parameter $\alpha$, that for $\alpha = 1$ also includes the KL divergence, which we consider in this work. However, the procedure proposed by Daudel et al. (2023) assumes $0 \le \alpha < 1$ and is thus not applicable in our setting. Morningstar et al. (2021) recently proposed a method for amortized variational inference with GMMs. In amortized VI, the parameters of the GMM are not optimized directly. Instead a neural network is trained to predict the GMM parameters based on a given input. NPVI (Gershman et al., 2012) was presented as nonparametric variational inference. However, it can also be regarded as a method for optimizing a very restricted type of GMM, namely, mixtures with uniform weights and Gaussian components with isotropic covariance matrices. Miller et al. (2017) and Guo et al. (2016) proposed boosting methods, which, however, can require unreasonably large mixture models, because components that have been added at previous iterations do not longer receive updates. Recently, Lin et al. (2021) investigated natural gradient GMM-based variational inference with a structured covariance matrix, which may be important for scaling these methods to higher dimensions.

## D  Background Material

We will now provide a brief introduction to natural gradient descent, Fisher information, and exponential family distributions.

### D.1  Natural Gradient Descent

A possible way to update the parameters $\boldsymbol{\theta}^{(i)}$ at iteration $i$ to minimize an objective $J(\boldsymbol{\theta})$ is the (vanilla) gradient descent update,

$$\boldsymbol{\theta}^{(i+1)} = \boldsymbol{\theta}^{(i)} - \beta \nabla_{\boldsymbol{\theta}} J(\boldsymbol{\theta}),$$

where $\beta$ is the stepsize. However, this update is not invariant to the chosen parameterization: Using a different parameterization $J_*(\boldsymbol{\lambda}^{(i)}) = J(\boldsymbol{\theta}^{(i)})$, performing the gradient descent update based on $\nabla_{\boldsymbol{\lambda}} J_*(\boldsymbol{\lambda})$ and changing the obtained parameters $\boldsymbol{\lambda}^{(i+1)}$ back to $\boldsymbol{\theta}^{(i+1)}$ may in general result in an update $\boldsymbol{\theta}^{(i)} \to \boldsymbol{\theta}^{(i+1)}$ that is not along the gradient descent in the original parameterization. To understand the reason for this dependency on the parameterization it helps to investigate the optimization problem that is solved by gradient descent. Namely—as can be easily verified by the Lagrangian method—, the gradient descent update maximizes a first-order Taylor approximation of the objective subject to a Euclidean trust region constraint,

$$\underset{\delta\boldsymbol{\theta}}{\arg\min} \quad J(\boldsymbol{\theta}^{(i)}) + \left( \nabla_{\boldsymbol{\theta}} J(\boldsymbol{\theta}) \Big|_{\boldsymbol{\theta}=\boldsymbol{\theta}^{(i)}} \right)^{\top} \delta\boldsymbol{\theta} \tag{10}$$

$$\text{s.t.} \quad \delta\boldsymbol{\theta}^{\top} \delta\boldsymbol{\theta} = \epsilon, \tag{11}$$

where the stepsize $\beta = (2\lambda)^{-1}$ depends on the Lagrangian multiplier $\lambda$ and, therefore on the trust region $\epsilon$. However, measuring the distance of the update step using the Euclidean distance of the parameters is in general not meaningful as it depends on the chosen parameterization.

Instead, natural gradient descent (Amari, 1998) uses more general Riemannian distances $d(\delta\boldsymbol{\theta}) = \delta\boldsymbol{\theta}^\top \mathbf{G}(\boldsymbol{\theta}^{(i)})\delta\boldsymbol{\theta}$, where $\mathbf{G}(\boldsymbol{\theta}^{(i)})$ is a matrix that may smoothly change as a function of $\boldsymbol{\theta}$, which is also called the Riemannian metric tensor. The Riemannian metric tensor defines a Riemannian manifold, a curved but locally flat manifold. Minimizing the first-order Taylor approximation subject to a trust region constraint on the Riemannian distance $d(\delta\boldsymbol{\theta})$ produces an update along the natural gradient

$$\tilde{\nabla}_{\boldsymbol{\theta}} J(\boldsymbol{\theta})\Big|_{\boldsymbol{\theta}=\boldsymbol{\theta}^{(i)}} = \mathbf{G}(\boldsymbol{\theta}^{(i)})^{-1}\nabla_{\boldsymbol{\theta}} J(\boldsymbol{\theta})\Big|_{\boldsymbol{\theta}=\boldsymbol{\theta}^{(i)}}$$

where the stepsize is again dependent on the trust region. Hence, the natural gradient can be computed by preconditioning the vanilla gradient with the inverse Riemannian metric tensor.

When optimizing an objective with respect to a probability distribution, we can use natural gradient descent based on a Riemannian distance that approximates the Kullback-Leibler divergence $\text{KL}(p_{\boldsymbol{\theta}^{(i)}}||p_{\boldsymbol{\theta}^{(i+1)}})$ between the old and the updated distribution. Such distance metric is typically much more appropriate than the Euclidean distance of the parameters, as it is invariant with respect to the chosen parameterization. The corresponding Riemannian metric tensor corresponds to the Fisher information matrix, which we will discuss in the next subsection.

## D.2   Fisher Information

The Fisher information matrix $F(\boldsymbol{\theta})$ can be derived as the Riemannian distance metric that is obtained from the second-order Taylor approximation of the KL divergence at $\delta\boldsymbol{\theta} = \mathbf{0}$,

$$\text{KL}(p_{\boldsymbol{\theta}}||p_{\boldsymbol{\theta}+\delta\boldsymbol{\theta}}) \approx \underbrace{\int_{\mathbf{x}} p_{\boldsymbol{\theta}}(\mathbf{x}) \log \frac{p_{\boldsymbol{\theta}}(\mathbf{x})}{p_{\boldsymbol{\theta}}(\mathbf{x})}d\mathbf{x}}_{=0} - \delta\boldsymbol{\theta}^\top \left[\int_{\mathbf{x}} p_{\boldsymbol{\theta}}(\mathbf{x})\nabla_{\boldsymbol{\theta}} \log p_{\boldsymbol{\theta}}(\mathbf{x})d\mathbf{x}\right] - \frac{1}{2}\delta\boldsymbol{\theta}^\top \left[\int_{\mathbf{x}} p_{\boldsymbol{\theta}}(\mathbf{x})\nabla_{\boldsymbol{\theta}\boldsymbol{\theta}} \log p_{\boldsymbol{\theta}}(\mathbf{x})d\mathbf{x}\right]\delta\boldsymbol{\theta}$$

$$= \underbrace{-\delta\boldsymbol{\theta}^\top \int_{\mathbf{x}} \nabla_{\boldsymbol{\theta}} p_{\boldsymbol{\theta}}(\mathbf{x})d\mathbf{x}}_{=0} - \frac{1}{2}\delta\boldsymbol{\theta}^\top \left[\int_{\mathbf{x}} p_{\boldsymbol{\theta}}(\mathbf{x})\nabla_{\boldsymbol{\theta}\boldsymbol{\theta}} \log p_{\boldsymbol{\theta}}(\mathbf{x})d\mathbf{x}\right]\delta\boldsymbol{\theta}$$

$$= \frac{1}{2}\delta\boldsymbol{\theta}^\top \left[-\int_{\mathbf{x}} p_{\boldsymbol{\theta}}(\mathbf{x})\nabla_{\boldsymbol{\theta}\boldsymbol{\theta}} \log p_{\boldsymbol{\theta}}(\mathbf{x})d\mathbf{x}\right]\delta\boldsymbol{\theta}$$

$$= \frac{1}{2}\delta\boldsymbol{\theta}^\top \underbrace{\left[\int_{\mathbf{x}} p_{\boldsymbol{\theta}}(\mathbf{x})\left(\nabla_{\boldsymbol{\theta}} \log p_{\boldsymbol{\theta}}(\mathbf{x})\right)\left(\nabla_{\boldsymbol{\theta}} \log p_{\boldsymbol{\theta}}(\mathbf{x})\right)^\top d\mathbf{x}\right]}_{F(\boldsymbol{\theta})}\delta\boldsymbol{\theta}.$$

Hence, the steepest descent update subject to a trust-region constraint based on the approximated KL divergence is given by the natural gradient

$$\tilde{\nabla}_{\boldsymbol{\theta}} J(\boldsymbol{\theta})\Big|_{\boldsymbol{\theta}=\boldsymbol{\theta}^{(i)}} = \mathbf{F}(\boldsymbol{\theta}^{(i)})^{-1}\nabla_{\boldsymbol{\theta}} J(\boldsymbol{\theta})\Big|_{\boldsymbol{\theta}=\boldsymbol{\theta}^{(i)}}.$$

We will now revisit exponential family distributions, a family of distributions for which the natural gradient can be computed without explicitly computing the Fisher information matrix, as mentioned in Section 3.2.

## D.3   Exponential Family Distributions

Exponential family distributions are distributions that can be written in the form

$$p(\mathbf{x}; \boldsymbol{\theta}) = h(\mathbf{x}) \exp\Big(\eta(\boldsymbol{\theta}) \cdot \mathbf{T}(\mathbf{x}) - A(\boldsymbol{\theta})\Big),$$

with base measure $h(\mathbf{x})$, natural parameters $\boldsymbol{\eta}(\boldsymbol{\theta})$, sufficient statistics $\mathbf{T}(\mathbf{x})$ and log partition function $A(\boldsymbol{\theta})$. Furthermore, exponential family distributions can be also specified using the expectation parameters, that is, the expected value of the sufficient statistics $\mathbf{m}(\boldsymbol{\theta}) = \int_{\mathbf{x}} p_{\boldsymbol{\theta}}(\mathbf{x})\mathbf{T}(\mathbf{x})d\mathbf{x}$.

Exponential family distributions include a wide range of distributions, such as $k$-dimensional Gaussian distributions, where

$$h(\mathbf{x}) = (2\pi)^{-\frac{k}{2}}, \quad \boldsymbol{\eta}(\boldsymbol{\theta}) = \begin{bmatrix} \boldsymbol{\Sigma}^{-1}\boldsymbol{\mu} \\ -\frac{1}{2}\boldsymbol{\Sigma}^{-1} \end{bmatrix}, \quad \mathbf{T}(\mathbf{x}) = \begin{bmatrix} \mathbf{x} \\ \mathbf{x}\mathbf{x}^\top \end{bmatrix}, \quad A(\boldsymbol{\theta}) = \frac{1}{2}\boldsymbol{\mu}^\top\boldsymbol{\Sigma}^{-1}\boldsymbol{\mu} + \frac{1}{2}\log|\boldsymbol{\Sigma}|$$

and categorical distributions, where

$$h(\mathbf{x}) = 1, \quad \boldsymbol{\eta}(\boldsymbol{\theta}) = \begin{bmatrix} \log\frac{p_1}{p_K} \\ \vdots \\ \log\frac{p_{K-1}}{p_K} \\ 0 \end{bmatrix}, \quad \mathbf{T}(\mathbf{x}) = \begin{bmatrix} [\mathbf{x} = 1] \\ [\mathbf{x} = 2] \\ \vdots \\ [\mathbf{x} = K] \end{bmatrix}, \quad A(\boldsymbol{\theta}) = -\log\left(1 - \sum_{i=1}^{K-1} p_i\right), \tag{12}$$

and where $\mathbf{T}(\mathbf{x})$ is the one-hot encoding of the outcome $\mathbf{x}$. For minimal exponential family distributions (where the natural parameters and sufficient statistics are linearly independent) the natural gradient with respect to the natural parameters coincides with the gradient with respect to the expectation parameters, that is

$$\mathbf{F}(\boldsymbol{\theta})^{-1}\nabla_{\eta(\boldsymbol{\theta})}J(\eta(\boldsymbol{\theta})) = \nabla_{\mathbf{m}(\boldsymbol{\theta})}J_*(\mathbf{m}(\boldsymbol{\theta}))$$

as shown by Khan and Nielsen (2018, c.f. Theorem 1).

## E   Proof of Theorem 2.1

The E-step of VIPS is performed by setting the auxiliary distribution to the current model. Using $\boldsymbol{\theta}^{(i)}$ to denote the model parameters at iteration $i$, the auxiliary distribution at iteration $i$ is given by

$$\tilde{q}(o|\mathbf{x}) = q_{\boldsymbol{\theta}^{(i)}}(o|\mathbf{x})$$

by construction. Following Becker et al. (2019) we express the auxiliary distribution in terms of $\tilde{q}(o) = q_{\boldsymbol{\theta}^{(i)}}(o)$, $\tilde{q}(\mathbf{x}) = q_{\boldsymbol{\theta}^{(i)}}(\mathbf{x})$, and $\tilde{q}(\mathbf{x}|o) = q_{\boldsymbol{\theta}^{(i)}}(\mathbf{x}|o)$ using $\tilde{q}(o|\mathbf{x}) = \frac{\tilde{q}(o)\tilde{q}(\mathbf{x}|o)}{\tilde{q}(\mathbf{x})}$, to reformulate the lower bound objective as,

$$\tilde{J}(\tilde{q}, \boldsymbol{\theta}) = \sum_o q_{\boldsymbol{\theta}}(o)\left[\int_{\mathbf{x}} q_{\boldsymbol{\theta}}(\mathbf{x}|o)\left(\log\frac{\tilde{p}(\mathbf{x})}{\tilde{q}(\mathbf{x})}\right)d\mathbf{x} - \mathrm{KL}(q_{\boldsymbol{\theta}}(\mathbf{x}|o)||\tilde{q}(\mathbf{x}|o)))\right] - \mathrm{KL}(q_{\boldsymbol{\theta}}(o)||\tilde{q}(o)), \tag{13}$$

$$\neq \sum_o q_{\boldsymbol{\theta}}(o)\int_{\mathbf{x}} q_{\boldsymbol{\theta}}(\mathbf{x}|o)\left(\log\frac{\tilde{p}(\mathbf{x})}{q_{\boldsymbol{\theta}}(\mathbf{x})}\right)d\mathbf{x} = J(\boldsymbol{\theta}). \tag{14}$$

We can see that the lower bound objective (Arenz et al., 2018; Becker et al., 2019) differs from the original ELBO objective by two additional KL divergences, and further, by computing the entropy based on the fixed auxiliary distribution $\tilde{q}(\mathbf{x}) = q_{\boldsymbol{\theta}^{(i)}}(\mathbf{x})$ rather than the model $q_{\boldsymbol{\theta}}(\mathbf{x})$. Directly after the E-Step, the lower bound is tight, $\tilde{J}(q_{\boldsymbol{\theta}^{(i)}}, \boldsymbol{\theta}^{(i)}) = J(\boldsymbol{\theta}^{(i)})$, because the auxiliary distributions are equal to the current model. For proving Theorem 2.1 we need to prove that also the gradients match, that is,

$$\nabla_{\boldsymbol{\theta}}\tilde{J}(q_{\boldsymbol{\theta}^{(i)}}, \boldsymbol{\theta})\Big|_{\boldsymbol{\theta}=\boldsymbol{\theta}^{(i)}} = \nabla_{\boldsymbol{\theta}}J(\boldsymbol{\theta})\Big|_{\boldsymbol{\theta}=\boldsymbol{\theta}^{(i)}}, \tag{15}$$

which would also imply that the natural gradients match, since the Fisher Information Matrix only depends on the current distribution $q_{\boldsymbol{\theta}}$, not on the objective.

For proving the equivalence of the gradients, the following lemma will become handy:

**Lemma E.1.** *The gradient of the Kullback-Leibler divergence between two equal distributions $q_{\boldsymbol{\theta}'}(\mathbf{x}) = \tilde{q}(\mathbf{x})$ is zero, that is,*

$$q_{\boldsymbol{\theta}'}(\mathbf{x}) = \tilde{q}(\mathbf{x}) \implies \nabla_{\boldsymbol{\theta}}KL(q_{\boldsymbol{\theta}}||\tilde{q})\Big|_{\boldsymbol{\theta}=\boldsymbol{\theta}'} = \mathbf{0}.$$

*Proof.*

$$\nabla_{\boldsymbol{\theta}} \mathrm{KL}(q_{\boldsymbol{\theta}} || \tilde{q}) = \nabla_{\boldsymbol{\theta}} \int_{\mathbf{x}} q_{\boldsymbol{\theta}}(\mathbf{x}) \log \frac{q_{\boldsymbol{\theta}}(\mathbf{x})}{\tilde{q}(\mathbf{x})} d\mathbf{x} = \int_{\mathbf{x}} \nabla_{\boldsymbol{\theta}} q_{\boldsymbol{\theta}}(\mathbf{x}) \cdot \log \frac{q_{\boldsymbol{\theta}}(\mathbf{x})}{\tilde{q}(\mathbf{x})} d\mathbf{x} + \int_{\mathbf{x}} \nabla_{\boldsymbol{\theta}} q_{\boldsymbol{\theta}}(\mathbf{x}) d\mathbf{x}$$
$$\underset{q_{\boldsymbol{\theta}} = \tilde{q}}{=} \int_{\mathbf{x}} \nabla_{\boldsymbol{\theta}} q_{\boldsymbol{\theta}}(\mathbf{x}) \cdot 0 \; d\mathbf{x} + \nabla_{\boldsymbol{\theta}} 1 = \mathbf{0}$$

$\square$

As the gradient of the KL divergence between two equal distributions is zero, these terms do not affect the gradient of the objective function directly after the E-step. Hence, we only need to show that

$$\nabla_{\boldsymbol{\theta}} \left[ \int_{\mathbf{x}} q_{\boldsymbol{\theta}}(\mathbf{x}) \log q_{\boldsymbol{\theta}}(\mathbf{x}) d\mathbf{x} \right] \Bigg|_{\boldsymbol{\theta} = \boldsymbol{\theta}'} = \nabla_{\boldsymbol{\theta}} \left[ \int_{\mathbf{x}} q_{\boldsymbol{\theta}}(\mathbf{x}) \log q_{\boldsymbol{\theta}^{(i)}}(\mathbf{x}) d\mathbf{x} \right] \Bigg|_{\boldsymbol{\theta} = \boldsymbol{\theta}'}.$$

This equivalence can be verified as follows:

$$\nabla_{\boldsymbol{\theta}} \left[ \int_{\mathbf{x}} q_{\boldsymbol{\theta}}(\mathbf{x}) \log q_{\boldsymbol{\theta}}(\mathbf{x}) d\mathbf{x} \right] = \int_{\mathbf{x}} \nabla_{\boldsymbol{\theta}} q_{\boldsymbol{\theta}}(\mathbf{x}) \cdot \log q_{\boldsymbol{\theta}}(\mathbf{x}) d\mathbf{x} + \int_{\mathbf{x}} \nabla_{\boldsymbol{\theta}} q_{\boldsymbol{\theta}}(\mathbf{x}) d\mathbf{x}$$
$$= \int_{\mathbf{x}} \nabla_{\boldsymbol{\theta}} q_{\boldsymbol{\theta}}(\mathbf{x}) \cdot \log q_{\boldsymbol{\theta}}(\mathbf{x}) d\mathbf{x} + \nabla_{\boldsymbol{\theta}} 1 \underset{\boldsymbol{\theta} = \boldsymbol{\theta}^{(i)}}{=} \int_{\mathbf{x}} \nabla_{\boldsymbol{\theta}} q_{\boldsymbol{\theta}}(\mathbf{x}) \cdot \log q_{\boldsymbol{\theta}^{(i)}}(\mathbf{x}) d\mathbf{x} = \nabla_{\boldsymbol{\theta}} \left[ \int_{\mathbf{x}} q_{\boldsymbol{\theta}}(\mathbf{x}) \log q_{\boldsymbol{\theta}^{(i)}}(\mathbf{x}) d\mathbf{x} \right].$$

Pretending for a moment that we are optimizing both objectives using the reparameterization trick, it is inspiring to relate the lower bound, to the common practice of not backpropagating through the score function, yielding an unbiased, but often lower-variance estimate of the ELBO gradient (Roeder et al., 2017).

## F  Equivalence of the Weight Updates

We will now compare the weight update used by Lin et al. (2019a; 2020), with the weight updated used by Arenz et al. (2018; 2020), and show their equivalence.

### F.1  Weight Update of Lin et al. (2019a; 2020)

Lin et al. (2020) formulate the weight update in the natural parameter space $\boldsymbol{\eta}$ (see Eq. 12),

$$\boldsymbol{\eta} = \boldsymbol{\eta}_{\mathrm{old}} + \beta \boldsymbol{\delta} \tag{16}$$

where each dimension $\boldsymbol{\delta}_o$ of the natural gradient is given by

$$\boldsymbol{\delta}_o = \mathbb{E}_q(\mathbf{x}|o) \left[ \log \tilde{p}(\mathbf{x}) - \log q_{\boldsymbol{\theta}}(\mathbf{x}) \right]$$
$$- \mathbb{E}_q(\mathbf{x}|K) \left[ \log \tilde{p}(\mathbf{x}) - \log q_{\boldsymbol{\theta}}(\mathbf{x}) \right].$$

Defining

$$\hat{R}(o) = \mathbb{E}_q(\mathbf{x}|o) \left[ \log \tilde{p}(\mathbf{x}) - \log q_{\boldsymbol{\theta}}(\mathbf{x}) \right],$$

the $i$-th index of the natural parameters is given by

$$\boldsymbol{\eta}_i = \boldsymbol{\eta}_{\mathrm{old},i} + \beta \hat{R}(o) - \beta \hat{R}(K).$$

The new weight of component $o < K$ is given by

$$q(o; \boldsymbol{\eta}) = \frac{\exp(\boldsymbol{\eta}_o)}{\left[\sum_{k=1}^{K-1} \exp \boldsymbol{\eta}_k\right] + 1} = \frac{\exp\left(\boldsymbol{\eta}_{\mathrm{old},o} + \beta \hat{R}(o) - \beta \hat{R}(K)\right)}{\left[\sum_{k=1}^{K-1} \exp\left(\boldsymbol{\eta}_{\mathrm{old},k} + \beta \hat{R}(k) - \beta \hat{R}(K)\right)\right] + 1}$$

$$= \frac{\frac{q(o)_{\mathrm{old}}}{q(K)_{\mathrm{old}}} \exp\left(\beta \hat{R}(o)\right)}{\left[\sum_{k=1}^{K-1} \frac{q(k)_{\mathrm{old}}}{q(K)_{\mathrm{old}}} \exp\left(\beta \hat{R}(k)\right)\right] + \exp \beta \hat{R}(K)} = \frac{q(o)_{\mathrm{old}} \exp\left(\beta \hat{R}(o)\right)}{\sum_{k=1}^{K} q(k)_{\mathrm{old}} \exp\left(\beta \hat{R}(k)\right)}.$$

Hence, for all components (including K), we have

$$q(o) \propto q(o)_{\mathrm{old}} \exp\left(\beta \hat{R}(o)\right). \tag{17}$$

## F.2 Weight Update (VIPS)

We start be expressing the component's reward $R(o)$ in terms of $\hat{R}(o)$, namely,

$$R(o) = \mathbb{E}_{q_{\boldsymbol{\theta}}(\mathbf{x}|o)}\left[\log \tilde{p}(\mathbf{x}) + \log \tilde{q}(o|\mathbf{x})\right] + \mathrm{H}(q(\mathbf{x}|o))$$

$$= \mathbb{E}_{q_{\boldsymbol{\theta}}(\mathbf{x}|o)}\left[\log \tilde{p}(\mathbf{x}) + \log q(o|\mathbf{x}) - \log q(\mathbf{x}|o)\right]$$

$$= \mathbb{E}_{q_{\boldsymbol{\theta}}(\mathbf{x}|o)}\left[\log \tilde{p}(\mathbf{x}) + \log q(o) - \log q(x)\right] = \hat{R}(o) + \log q(o),$$

where we exploited that the auxiliary distribution is chosen according to the true responsibilities, $\tilde{q}(o|x) = q(o|x)$.

Expressing the weight update of VIPS (Arenz et al., 2018, Eq.8) in terms of $\hat{R}(o)$,

$$q(o) \propto \left[q_{\mathrm{old}}(o)^{\frac{\eta_w}{1+\eta_w}} \exp\left(R(o)\right)^{\frac{1}{1+\eta_w}}\right.$$

$$= q_{\mathrm{old}}(o)^{\frac{\eta_w}{1+\eta_w}} \exp\left(\hat{R}(o) + \log q_{\mathrm{old}}(o)\right)^{\frac{1}{1+\eta_w}}$$

$$\left. = q_{\mathrm{old}}(o) \exp\left(\frac{1}{1+\eta_w} \hat{R}(o)\right)\right],$$

we can see that it exactly matches the update in Equation 17 for $\beta = \frac{1}{1+\eta_w}$. □

## G Comparisons with Reference Implementations

The target distributions *BreastCancer*, *GermanCredit*, *GMM20* and *PlanarRobot4* were taken from VIPS (Arenz et al., 2020). The design choices of VIPS correspond to the codename ZAMTRUX in our implementation. Table 6 compares the final (negated) ELBO that we achieved in our main experiments (cf. Table 5) with the performance of VIPS reported by Arenz et al. (2020), that was obtained using their (C++) implementation. For all environments that have been tested in both works, the final ELBO performances published in this work are better than the results published in the original work, except for *GermanCredit* were, we could not measure any difference between both implementations.

We also compared our implementation with the Matlab implementation of Lin et al. (2020) on the target distributions that we took from their work (*STM20* and *STM300*). However, our problem setting slightly differs from the original setting, as Lin et al. (2020) use an expensive Hessian-based pre-training for initializing the GMM and compare the methods only with respect to their fine-tuning performance. We found that such pre-training is in general not necessary with our implementation and directly use the respective algorithms

| Implementation | BreastCancer | GermanCredit | GMM20 | PlanarRobot4 |
|---|---|---|---|---|
| VIPS/ZAMTRUX (theirs) | $78.20 \pm 0.04$ | $585.10 \pm 0.00$ | $0.01 \pm 0.00$ | $12.02 \pm 0.13$ |
| VIPS/ZAMTRUX (ours) | $\mathbf{78.14 \pm 0.01}$ | $585.10 \pm 0.00$ | $\mathbf{-0.00 \pm 0.00}$ | $\mathbf{11.48 \pm 0.04}$ |

Table 6: We compare the final (negated) ELBO achieved by both implementations. When using the same design choices as VIPS (Arenz et al., 2020), our implementation and hyperparameters led to better approximations on their target distributions.

starting from the original initialization of Lin et al. (2020). Figure 1, which compares the learned model and the target distributions based on the marginals on the *STM20* experiment, demonstrates that even without pre-training, we can learn higher-quality approximations with our implementation (cf. Lin et al., 2020, Fig.3). Here, we used the SAMTRON design choices, which performed best in our experiments.

We ran the STM20 experiments on the reference implementation (with disabled pre-training) using our hyperparameters, and using the hyperparameters of Lin et al. (2020) and compare the learning curves with our SEPYFUX evaluation in Figure 2a. Our hyperparameters achieve better final ELBO even on the original implementation. Using the same hyperparameters, the learning curves of both implementations do not differ significantly, but our implementation performed slightly better. The *STM300* experiment was not evaluated by Lin et al. (2020) in the first-order setting, but only when using Hessian information (using Eq. 6 left), which is often computationally prohibitive. Hence, we can only compare both implementations using our hyperparameters. The respective learning curves are shown in Figure 2b and very similar for both implementations.

## H   Hyperparameters

We list the hyperparameters for each design choice in Table 7. Please refer to Appendix I for a description of the different hyperparameters. The tested and eventually chosen hyperparameters for each experiment can be found in the reproducibility package; please refer to its `README.rst` file for links to all relevant config files.

## I   Notes On Our Implementations

We implemented two different options for estimating the natural gradients for the component update in two separate classes. The *MoreNgEstimator* uses weighted least squares to estimate the natural gradient using compatible function approximation; the *SteinNgEstimator* makes use of gradient information, using Stein's Lemma to estimate the natural gradient. For both options, a boolean hyperparameter is available to select whether self-normalized importance weighting, or standard importance weighting should be used to make use of samples from different distributions. Furthermore, a boolean hyperparameter can be used to disable importance sampling, only using samples from the respective component during the component update, which we enabled for all methods on *WINE* to reduce memory footprint. When using MORE, an additional hyperparameter can be used to select the initial $\ell_2$-regularization for linear least-squares; the regularizer is automatically adapted as described by Arenz et al. (2020).

We implemented three options for performing the natural gradient update of the components based on the estimated natural gradients and given stepsizes (or trust regions). The *DirectNgBasedComponentUpdater* directly applies the natural gradient update given by Equation 4. If an updated component is no longer positive definite, the update is undone, hoping that the update would succeed in the next iteration (potentially with a smaller stepsize). The *NgBasedComponentUpdaterIblr* updates the components according to the improved Bayesian learning rule (Eq. 8). We noticed that the update may still result in non-invertible covariance matrices (albeit much less frequently compared to the natural gradient update) due to numerical errors, in which case we also undo the respective updates. The *KLConstrainedNgBasedComponentUpdater* solves an optimization problem to find stepsizes that result in positive definite covariance matrices and updated distributions that respect the desired bound on the KL divergence, as discussed in Section 3.3. For solving the convex optimization problem, Arenz et al. (2018) used an L-BFGS-B (Byrd et al., 1995)

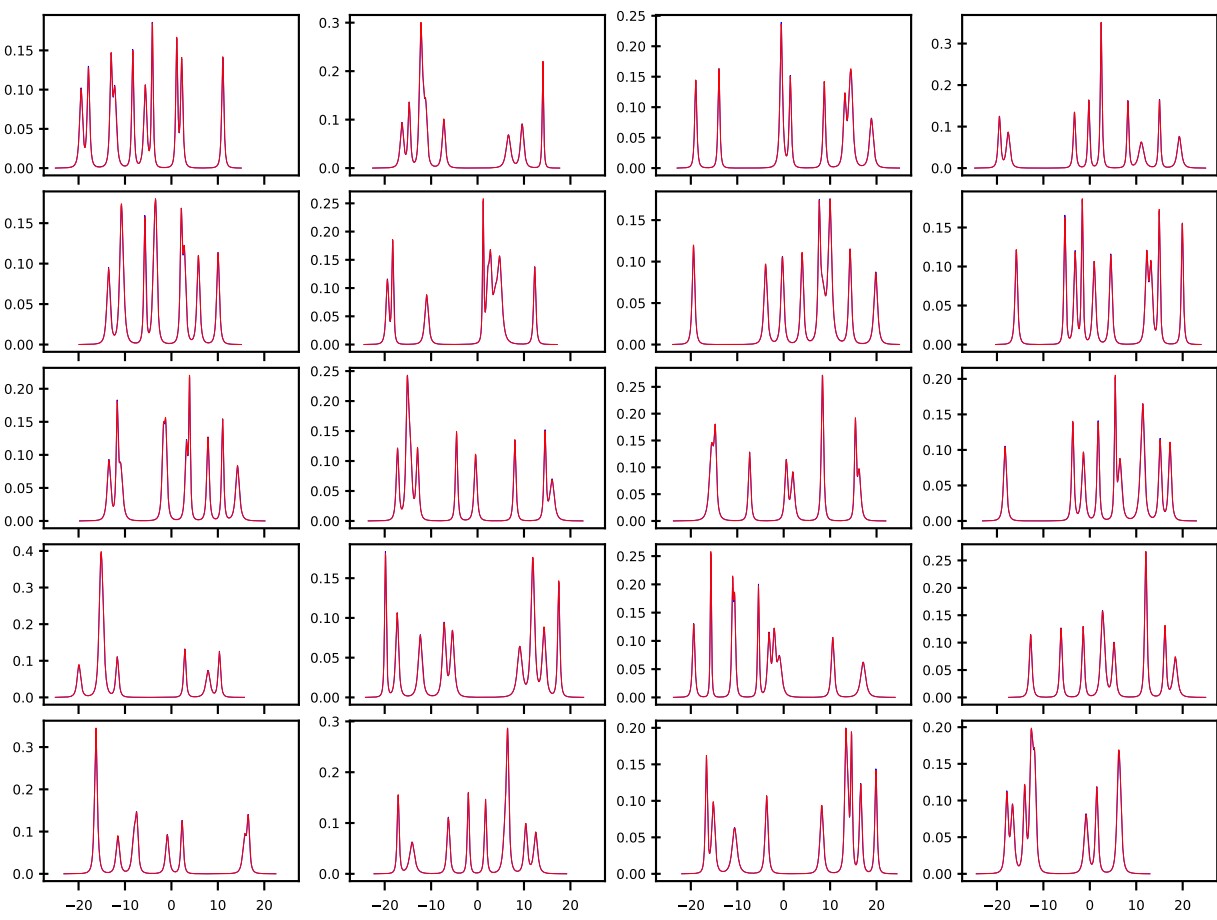

Figure 1: A representative plot of the 20 marginal distributions of the GMM learned with SAMTRON for the *STM20* experiment is shown in red. The marginals of the Mixture of Student-T target distribution are shown in blue and hardly distinguishable.

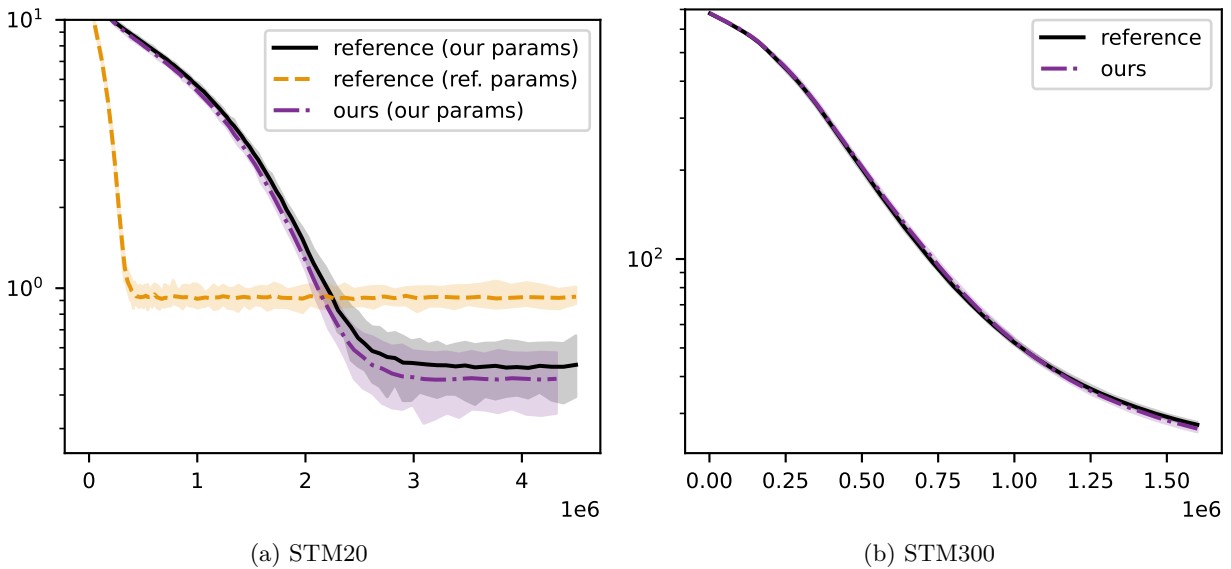

(a) STM20                                    (b) STM300

Figure 2: The learning curves plot the ELBO (in logarithmic scale) over the number of samples for our implementation and the reference implementation, where both use the SEPYFUX design choices. On *STM20* our hyperparameters lead to a better approximation than the hyperparameters used by Lin et al. (2020), and when using the same hyperparameters, our implementation performs slightly better. On *STM300* both implementations perform similarly using our hyperparameters. Reference hyperparameters are not available for *STM300*.

| Module | Design Choice | Hyperparameters | | | |
|---|---|---|---|---|---|
| Component Adaptation | Fixed Adaptive | - adding iter. | - deletion iter. | - # prior samples | - # DB samples |
| Component Stepsize Adaptation | Fixed Decaying Adaptive | initial_stepsize initial stepsize initial stepsize | - annealing exponent min stepsize | - - max stepsize | - - - |
| NgBased Component Updater | Direct iBLR Trust-Region | - - - | - - - | - - - | - - - |
| NgEstimator | Stein MORE | self-normalized IW self-normalized IW | use own samples use own samples | - $\ell_2$-coefficient | - - |
| Sample Selector | Comp.-Based Mixture-Based | desired samples desired samples | reused samples reused samples | - - | - - |
| Weight Stepsize Adaptation | Fixed Decaying Adaptive | initial_stepsize initial stepsize initial stepsize | - annealing exponent min stepsize | - - max stepsize | - - - |
| Weight Updater | Direct Trust-Region | self-normalized IW self-normalized IW | - - | - - | - - |

Table 7: The table lists each hyperparameter that was tuned at any of the experiments.

optimizer. However, it was difficult to efficiently integrate such optimizer into our Tensorflow (Abadi et al., 2015) implementation, and furthermore, the optimizer would sometimes fail for numerical reasons. Instead, we implemented a simple bisection method, by iteratively refining an initially sufficiently large bracket on $\log \beta_o$ by evaluating the center of the bracket and using it as new upper value when the corresponding KL divergence is too small or the covariance matrix not positive definite, or, otherwise, as new lower value. Although naive, this approach is numerically robust by only requiring the sign of the gradient of the KL divergence, converges to the optimum due to the convexity of the problem and can be efficiently compiled into a compute graph.

For the *SampleSelector* module, we implemented two options. Both options can make use of samples from previous iterations (as discussed by Arenz et al. (2020)), by setting the hyperparameter $n_{\text{reused}}$ larger than zero. The *LinSampleSelector* computes the effective sample size $n_{\text{eff}}$ on the GMM and draws $\max(0, N \cdot n_{\text{des}} - n_{\text{eff}})$ new samples from the GMM. For $n_{\text{reused}} = 0$ this approach corresponds to the procedure used by iBayes-GMM (Lin et al., 2020). The *VipsSampleSelector* computes $n_{\text{eff}}(o)$ for each component and draws $\max(0, n_{\text{des}} - n_{\text{eff}}(o))$ new samples from each component, matching the procedure used by VIPS (Arenz et al., 2020).

We implemented two options for performing the weight update. The *DirectWeightUpdater* directly updates the categorical distribution using Eq. 9, whereas the *TrustRegionBasedWeightUpdater* uses our bracketing search to stay within a given trust region. Whether standard importance sampling or self-normalized importance sampling should be used for estimating the component rewards $R(o)$ can be chosen with a hyperparameter, that is available for both options and can be chosen independently to the respective hyperparameter of the *NgEstimator*.

For stepsize adaptation, we implemented three options for both, the *ComponentStepsizeAdaptation* module and the *WeightStepsizeAdaptation* module. The *FixedComponentStepsizeAdaptation* and *FixedWeightStepsizeAdaptation* simply return a fixed stepsize, chosen as a hyperparameter. The *DecayingComponentStepsizeAdaptation* and *DecayingWeightStepsizeAdaptation* return an exponentially decaying stepsize based on the number of times the respective distribution has been updated, as described by Khan et al. (2018). The *ImprovementBasedComponentStepsizeAdaptation* and *ImprovementBasedWeightStepsizeAdaptation* increase the stepsize if the last update of the respective distribution improved its reward, and decrease it otherwise, as proposed by Arenz et al. (2020).

For the *ComponentAdaptation* module, we implemented two options. The *FixedComponentAdaptation* does nothing, keeping the number of components fixed throughout optimization; the *VipsComponentAdaptation* uses the procedure of VIPS (Arenz et al., 2020) to delete bad components, and to add new components in promising regions. Arenz et al. (2020) only considered samples from the sample database for initializing the mean of the new component, however, we had to disable the sample database for the high-dimensional problems *STM300* and *WINE* to reduce memory footprint. Hence, we introduced an additional hyperparameter to specify additional samples from the prior distribution (the same distribution that was used for drawing the means of the initial GMM), that should be drawn specifically for *ComponentAdaptation*.

## J  Test Problems

We performed several experiments on target distributions taken from related work. For additional details, please refer to the corresponding work. The *BreastCancer*, *GermanCredit*, *PlanarRobot* and *GMM* experiments were used by Arenz et al. (2020) and the Student-T experiments were used by Lin et al. (2020). For the most detailed (and fully accurate) specification of all target distributions, please refer to our code supplements (https://github.com/OlegArenz/gmmvi).

- In the *BreastCancer* and *GermanCredit* experiments, we aim to approximate the posterior distribution of a logistic regression problem. The dimensions are 25 and 31, respectively. The data sets can be obtained from the UCI Machine Learning Repository (Lichman, 2013) and contain 1000 and 569 data points. Mimicking the setup of Arenz et al. (2020), the ELBO is computed on the full data set during training and evaluation. For the minibatch variants *GermanCreditMB* and *BreastCancerMB* we also use batches from the full data set, such that the respective ELBOs are comparable with the

original experiments. However, we report the full-batch ELBOs to remove unnecessary noise in the evaluation.

- In the *WINE* experiment we want to approximate the posterior distribution over the weights of a neural network that predics the scalar wine quality based on eleven features using the WINE data set Lichman (2013). The network has two hidden layers of width 8, resulting in 177 parameters. The likelihood is given by the root mean squared error over a minibatch of size 128. We split the data set into a training and test set, and make sure that the training and test sets are deterministic given the seed. As we are primarily interested in the ability of the different methods to approximate a given target distribution (rather than testing how the approximations perform on the downstream task), we report the training set ELBOs. However, Table 8 reports as secondary metric the mean squared error of the prediction using approximate Bayesian inference on the test set.

- In the planar robot experiment, we aim to sample joint configurations of a 10-link *planar robot* (all links have the same length) and aim to reach one of four goal positions. The target distribution is Gaussian in the endeffector configuration space (but non-Gaussian in the joint configuration space). A zero-mean Gaussian prior on the joint angles is additionally used to prevent non-smooth configuration.

- In the *GMM* experiment we aim to approximate an unknown Gaussian mixture model with 10 components and a varying number of dimensions. Arenz et al. (2020) only considered 60 dimensions, but we increased the dimensionality to up to 100. The mean is sampled uniformly in the range $[-50, 50]$ and the covariance matrices $\mathbf{\Sigma} = \mathbf{A}^\top \mathbf{A} + I$ are created by randomly sampling the elements of the square matrix $\mathbf{A}$.

- The mixture of Student-T experiment (*STM*) is similar to the *GMM* experiment but uses Student-T components instead of Gaussians. We exactly follow Lin et al. (2020) by considering a 20-dimensional mixture with 10 components with mean uniformly sampled in $[-20, 20]$, and a 300-dimensional mixture with 20 components sampled in $[-25, 25]$. We follow Lin et al. (2020) by initializing the components of the GMM by sampling the mean from a zero-mean Gaussian distribution with diagonal covariance with standard deviation 100, and by initializing the diagonal covariance matrices with $\mathbf{\Sigma}_{\text{init}} = 300\mathbf{I}$. However, instead of pre-training with a second-order method, we directly start training from the initial GMM.

- The *TALOS* experiment is based on the implementation by Pignat et al. (2020). The poses of both feet, as well as the positions of the left end-effector and the center-of-mass are computed for the given joint positions based on a kinematic model of the robot. The target positions of the feet are given by $[-0.02, 0.09, -0.]$ and $[-0.02, -0.09, -0.]$ and their target orientation are given by identity rotation matrices $\mathbf{R} = \mathbf{I}_3$. The likelihood for each foot, is given by a 12-dimensional Gaussian distribution that penalizes deviations from these goal-parameters using a standard deviation of 0.2 for the Cartesian positions and a standard deviation of 0.1 for each entry of the rotation matrix. The likelihood of the left endeffector position is given by a Gaussian with mean $[0.1, 0.5, 1.]$ and diagonal standard deviations of 0.02. Violations of the inequality constraints that the joint angles should be within their limits, and the center-of-mass within the convex polygon spanned by the feet are penalized using the log-density of a Gaussian distribution placed on the violated bound (Pignat et al., 2020) using a standard deviation of 0.01 for the center-of-mass and 0.05 for the joint limits.

## K   Full Table for Experiment 3

The complete table for Experiment 3, showing all tested candidates, can be found in Table 8.

## L   Learning Curves

The learning curves (ELBO over time) for our main experiments are shown in Figure 3.

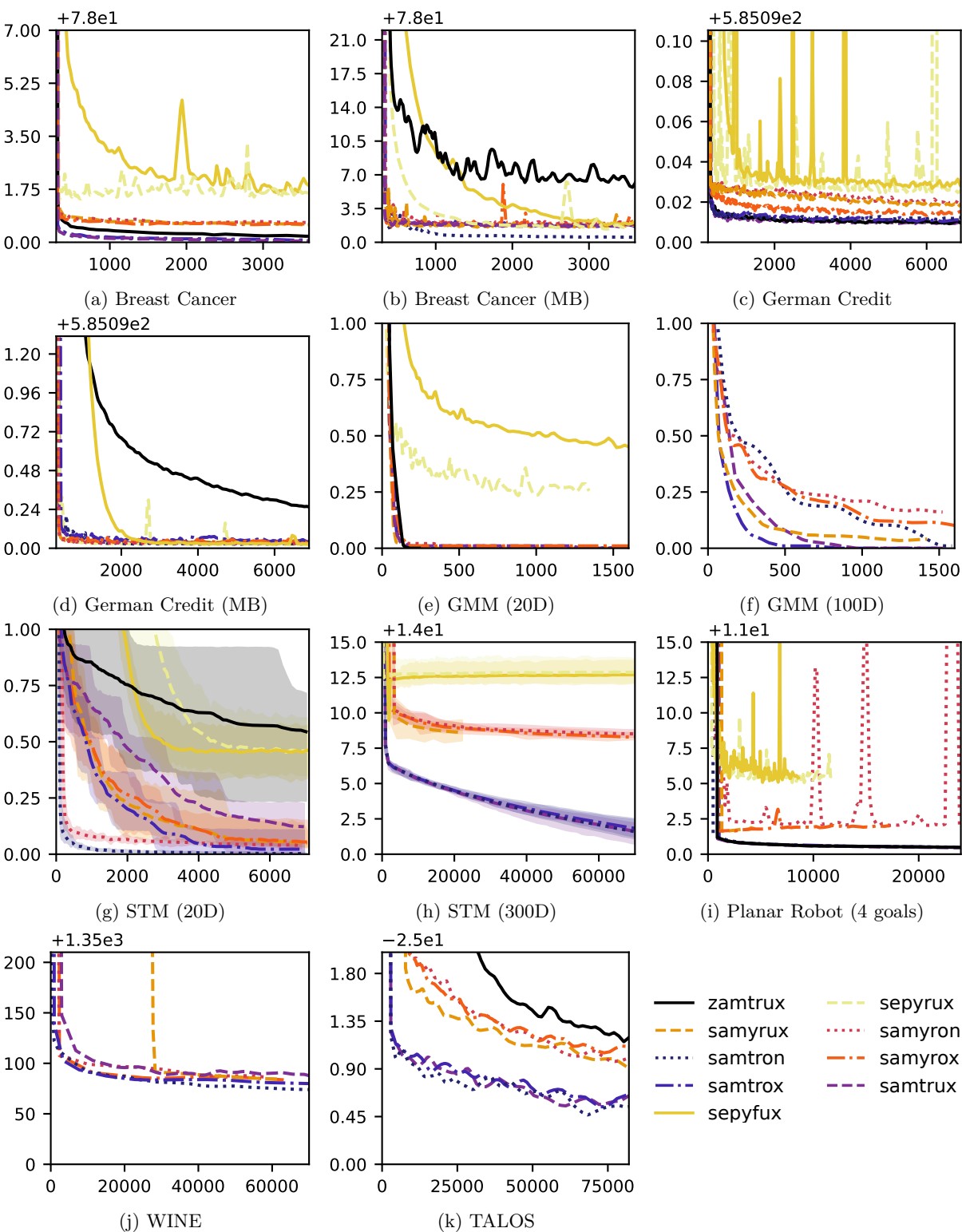

Figure 3: The learning curves for our main experiment (Experiment 3) show the negated ELBO over time (in seconds). Shaded areas show best and worst performance. Note that hyperparameters were selected with respect to final ELBO.

| Experiment | Metric | SAMTRUX | SAMTROX | SAMTRON | SAMYRUX | SAMYROX | SAMYRON | SEPYFUX | SEPYRUX | ZAMTRUX |
|---|---|---|---|---|---|---|---|---|---|---|
| BreastCancer | -ELBO | **78.01** ±**0.02** | 78.05 ±0.02 | **78.00** ±**0.02** | 78.57 ±0.07 | 78.63 ±0.07 | 78.61 ±0.05 | 79.78 ±0.40 | 79.91 ±0.93 | 78.14 ±0.01 |
| | MMD | **1.1e−3** ±**1e−4** | **1.1e−3** ±**1e−4** | **9.9e−4** ±**8e−5** | 1.2e−3 ±1e−4 | **1.1e−3** ±**2e−4** | 1.3e−3 ±1e−4 | 2.3e−3 ±3e−4 | 2.8e−3 ±3e−4 | **1.1e−3** ±**1e−4** |
| BreastCancer (minibatches) | -ELBO | 79.66 ±0.32 | 79.71 ±0.43 | **78.41** ±**0.03** | 79.68 ±0.19 | 79.94 ±0.23 | 79.96 ±0.26 | 80.13 ±0.69 | 79.38 ±0.25 | 83.65 ±0.97 |
| | MMD | **2.9e−3** ±**1e−3** | **2.6e−3** ±**8e−4** | **1.7e−3** ±**2e−4** | **2.2e−3** ±**8e−4** | 2.8e−3 ±9e−4 | 2.9e−3 ±8e−4 | 2.5e−3 ±4e−4 | **1.9e−3** ±**2e−4** | 8.3e−3 ±2e−3 |
| GermanCredit | -ELBO | **585.10** ±**0.00** | **585.10** ±**0.00** | **585.10** ±**0.00** | 585.11 ±0.00 | 585.10 ±0.00 | 585.11 ±0.01 | 585.12 ±0.01 | 585.11 ±0.01 | **585.10** ±**0.00** |
| | MMD | **5.5e−4** ±**4e−5** | **5.9e−4** ±**6e−5** | **5.5e−4** ±**5e−5** | **5.7e−4** ±**3e−5** | **5.5e−4** ±**4e−5** | **5.7e−4** ±**3e−5** | **5.6e−4** ±**5e−5** | **5.6e−4** ±**5e−5** | 6.0e−4 ±7e−5 |
| GermanCredit (minibatches) | -ELBO | **585.12** ±**0.01** | 585.13 ±0.01 | **585.12** ±**0.00** | 585.13 ±0.00 | 585.13 ±0.01 | **585.12** ±**0.00** | **585.12** ±**0.00** | **585.12** ±**0.00** | 585.35 ±0.03 |
| | MMD | **6.5e−4** ±**1e−4** | 8.1e−4 ±2e−4 | **5.9e−4** ±**4e−5** | 6.0e−4 ±5e−5 | 6.2e−4 ±7e−5 | 5.4e−4 ±5e−5 | 5.8e−4 ±1e−4 | 5.7e−4 ±6e−5 | 1.9e−3 ±6e−4 |
| Planar Robot | -ELBO | **11.47** ±**0.05** | **11.47** ±**0.04** | **11.47** ±**0.04** | 13.16 ±0.20 | 12.98 ±0.09 | 12.93 ±0.13 | 17.26 ±2.13 | 16.35 ±0.65 | **11.48** ±**0.04** |
| | MMD | **1.6e−2** ±**8e−4** | **1.6e−2** ±**2e−3** | **1.6e−2** ±**1e−3** | 3.7e−2 ±7e−3 | 2.6e−2 ±2e−3 | 2.4e−2 ±2e−3 | 4.5e−1 ±6e−2 | 4.4e−1 ±7e−2 | **1.6e−2** ±**9e−4** |
| GMM20 | -ELBO | −**0.00** ±**0.00** | **0.00** ±**0.00** | −**0.00** ±**0.00** | **0.00** ±**0.00** | **0.01** ±**0.03** | **0.00** ±**0.00** | 0.43 ±0.15 | 0.28 ±0.15 | −**0.00** ±**0.00** |
| | Modes | **10.00** ±**0.00** | **10.00** ±**0.00** | **10.00** ±**0.00** | **10.00** ±**0.00** | 9.90 ±0.28 | **10.00** ±**0.00** | 7.90 ±1.16 | 8.80 ±1.11 | **10.00** ±**0.00** |
| GMM100 | -ELBO | **0.00** ±**0.00** | **0.00** ±**0.00** | **0.01** ±**0.03** | 0.05 ±0.03 | 0.09 ±0.07 | 0.16 ±0.09 | 4.54 ±0.52 | 0.96 ±0.36 | 1.21 ±0.15 |
| | Modes | **10.00** ±**0.00** | **10.00** ±**0.00** | 9.90 ±0.28 | **10.00** ±**0.00** | 9.70 ±0.43 | 9.40 ±0.46 | 0.00 ±0.00 | 7.00 ±1.85 | 3.00 ±0.42 |
| STM20 | -ELBO | 0.12 ±0.07 | **0.02** ±**0.04** | **0.00** ±**0.00** | 0.04 ±0.02 | 0.05 ±0.03 | 0.03 ±0.01 | 0.46 ±0.06 | 0.46 ±0.07 | 0.54 ±0.17 |
| | Modes | 8.90 ±0.66 | 9.80 ±0.38 | **10.00** ±**0.00** | **10.00** ±**0.00** | 9.90 ±0.28 | **10.00** ±**0.00** | **9.20** ±**0.83** | 9.20 ±0.71 | 6.10 ±0.99 |
| STM300 | -ELBO | **15.00** ±**0.38** | 15.24 ±0.36 | 14.96 ±0.48 | 22.41 ±0.23 | 22.28 ±0.13 | 22.50 ±0.15 | 26.69 ±0.39 | 26.87 ±0.45 | N/A |
| | Modes | **13.70** ±**1.80** | 14.10 ±1.50 | 14.30 ±1.53 | 9.57 ±1.73 | 9.80 ±1.58 | 9.10 ±0.99 | 0.90 ±0.89 | 0.30 ±0.43 | N/A |
| WINE | -ELBO | **1435.65** ±**34.06** | **1429.02** ±**30.93** | **1423.12** ±**31.55** | **1432.72** ±**30.49** | **1430.80** ±**30.66** | **1433.87** ±**30.49** | 3279.32 ±1425.12 | 4100.92 ±1295.57 | 16503.38 ±813.75 |
| | MSE | **0.48** ±**0.02** | **0.48** ±**0.02** | **0.47** ±**0.02** | **0.48** ±**0.02** | **0.47** ±**0.02** | **0.48** ±**0.02** | 0.67 ±0.14 | 0.76 ±0.17 | 0.80 ±0.13 |
| TALOS | -ELBO | −**24.32** ±**0.22** | −**24.30** ±**0.10** | −**24.43** ±**0.16** | −**24.13** ±**0.14** | −23.91 ±0.13 | −24.00 ±0.23 | −16.64 ±5.26 | −19.00 ±1.07 | −23.69 ±0.16 |
| | $H(q)$ | −**16.81** ±**0.07** | −**16.88** ±**0.09** | −**16.82** ±**0.07** | −17.26 ±0.10 | −17.32 ±0.16 | −17.25 ±0.16 | −25.03 ±5.46 | −22.34 ±1.11 | −**16.91** ±**0.07** |

Table 8: The full table for our main experiment shows all tested candidates, as well as the secondary metrics.

| Experiment | Metric | *Samtrux* | *Samtrox* | *Samtron* | *Samyrux* | *Samyrox* | *Samyron* | *Sepyfux* | *Sepyrux* | *Zamtrux* |
|---|---|---|---|---|---|---|---|---|---|---|
| BreastCancer | -ELBO | **78.00** ±**0.02** | 78.04 ±0.02 | **78.01** ±**0.01** | 78.62 ±0.08 | 78.62 ±0.05 | 78.61 ±0.06 | 79.68 ±0.40 | 80.13 ±0.79 | 78.15 ±0.02 |
| | ACCURACY | **98.72** ±**0.11** | **98.79** ±**0.12** | **98.75** ±**0.09** | **98.79** ±**0.09** | **98.70** ±**0.08** | **98.70** ±**0.11** | **98.77** ±**0.13** | **98.75** ±**0.09** | **98.70** ±**0.11** |
| BreastCancer (minibatches) | -ELBO | 79.54 ±0.19 | 79.54 ±0.24 | **78.40** ±**0.03** | 79.70 ±0.20 | 79.90 ±0.22 | 79.95 ±0.23 | 79.83 ±0.70 | 79.59 ±0.70 | 85.88 ±3.41 |
| | ACCURACY | **98.73** ±**0.10** | **98.79** ±**0.12** | 98.66 ±0.08 | **98.72** ±**0.15** | **98.77** ±**0.07** | **98.79** ±**0.14** | **98.80** ±**0.10** | 98.75 ±0.05 | **98.89** ±**0.08** |
| GermanCredit | -ELBO | **585.10** ±**0.00** | **585.10** ±**0.00** | **585.10** ±**0.00** | 585.11 ±0.00 | 585.11 ±0.00 | 585.11 ±0.00 | 585.12 ±0.01 | 585.12 ±0.01 | **585.10** ±**0.00** |
| | ACCURACY | **78.47** ±**0.13** | **78.47** ±**0.10** | **78.55** ±**0.10** | **78.47** ±**0.10** | **78.55** ±**0.06** | **78.47** ±**0.10** | **78.52** ±**0.12** | **78.53** ±**0.09** | **78.54** ±**0.11** |
| GermanCredit (minibatches) | -ELBO | **585.12** ±**0.01** | 585.14 ±0.01 | **585.12** ±**0.00** | 585.13 ±0.00 | 585.12 ±0.01 | 585.12 ±0.00 | **585.12** ±**0.01** | **585.11** ±**0.00** | 585.32 ±0.02 |
| | ACCURACY | **78.52** ±**0.09** | **78.52** ±**0.15** | **78.55** ±**0.10** | 78.49 ±0.09 | 78.53 ±0.09 | 78.52 ±0.13 | 78.49 ±0.11 | **78.54** ±**0.10** | 78.46 ±0.14 |

Table 9: We reran the experiments on the logistic regression tasks where we also evaluated to accuracy of the Bayesian inference predictions (based on 2000 samples from the learned model). These experiments use the same hyperparameters and seeds that were used for the experiments reported in Table 8. For these tasks, slightly better approximations of the posterior, did not result in significant differences in the prediction accuracy.

## M Tips for the Practitioner

We do not want to make the impression that any variant, e.g. our proposed hybrid variant Samtron is to be unconditionally preferred. Instead, the different variants have their own advantages and disadvantages. In this section, we would like to discuss some tradeoffs that should be considered when applying our generalized framework to a practical problem.

### M.1 Exploitation vs. Exploration

We found that exploration is a key concern, when aiming to learn a highly accurate approximation that covers many different modes of the target distribution. Several design choices are particular important for discovering different modes.

- **Sampling from the Components.** Based on our experience, sampling from the individual components (design choice **M**) is not only important for discovering different modes, but also for ensuring that the discovered modes are actually covered by components with appropriate weight. Sampling from the mixture (**P**) may lead to a vicious circle, where components that don't approximate their respective mode very well, will have low weight and, thus, not receive sufficient samples to improve the approximation and obtain larger weight. On the other hand, design choice **M** may end up sampling at locations that are not relevant for a good approximation, which may lead to bad sample efficiency. Hence, when very fast learning is essential, **P** can be preferable.

- **Adapting the Number of Components.** Adapting the number of components by adding new components in promising regions is a very effective technique to improve exploration. Often, initializing a GMM with a single, high-entropy component and adding new components at good sample locations (Arenz et al., 2020) is still able to discover multiple different modes of the target distribu-

tion. However, we found that initializing with many components is an effective strategy to further improve exploration. As the component adaptation strategy also involves deleting components with low weights, it is even feasible to start with large initial GMMs and naturally reduce its size by deleting low weight components that no longer improve. Dynamically growing and shrinking the size of the GMM can not only help in exploration but may also improve sample efficiency, and therefore seems generally preferable compared to optimizing a fixed-size GMM.

- **Number of Samples.** The hyperparameter of the *SampleSelector* that specifies the number of desired samples does not only trade off training stability and efficiency, but also affects exploration. Hence, it may be beneficial to use relatively large sample sizes, even if they are not needed for improving the natural gradient estimates.

## M.2    Efficiency vs. Stability

The training stability is another major concern. However, when using adaptive trust regions, we can often obtain efficient and stable updates with little hyperparameter tuning.

- **Sample Size and Learning Rate.** The number of samples used for estimating the natural gradient and the step size (or trust region) of the natural gradient update should be considered in combination, since better estimates allow for larger step sizes. We found that adaptive trust regions for the component updates (design choices **T** and **R**) can ensure stable improvements for a wide range of different sample sizes, which results in good sample efficiency and stability with little hyperparameter tuning. However, too small step sizes can lead to computational overhead and too large step sizes can harm exploration. When evaluating the target distribution is cheap, we recommend to use large sample sizes while upper-bounding the maximum trust-region to help exploration.

- **Trust Regions.** While the performance of the iBLR update was often comparable to trust regions, it was typically at least slightly worse, and also the step size adaptation seems to work slightly better when controlling a trust region rather than the learning rate. Hence, we recommend trust regions for the component updates, although the iBLR update can also be worth trying. For the weight update, adaptive trust regions are preferably as they require little hyperparameter tuning. However, also directly controlling the stepsize, and even using fixed stepsize can achieve the same performance with good hyperparameters.

- **Natural Gradient Estimation.** Exploiting first-order information by using Stein's lemma for estimating the natural gradient (design choice **S**) is often around one order of magnitude more efficient than use the zero-order method MORE. Furthermore, solving the weighted least-squares problem can become relatively slow for large number of dimensions. Hence, we recommend to always use Stein's lemma, when the target distribution is differentiable.

