# OpenReview forum: "A Unified Perspective on Natural Gradient Variational Inference with Gaussian Mixture Models"
_TMLR — Accepted by TMLR_

### Review · Reviewer_dU7d · 2023-05-13

**Summary Of Contributions:**

The authors are interested in two algorithms for GMM-based variational inference, namely VIPS and iBayes-GMM. They obtain that these algorithms have equivalent derived updates, despite having different implementations and theoretical guarantees. They then evaluate empirically the impact of different implementation choices.

**Audience:**

Yes

**Broader Impact Concerns:**

I do not see any concerns on the ethical implications of the work.

**Claims And Evidence:**

No

**Requested Changes:**

Clarifying the manuscript according to the **1. Clarity** above is a critical adjustment for this work to be considered for publication. **2. Related work** would strengthen the work.

**Strengths And Weaknesses:**

I appreciate the effort that the authors have put into providing a unified perspective on natural gradient variational inference with mixture models. However, I think the clarity of the work should be improved for it to be considered for publication.

Please find below a detailed review regarding the aspects that I think could be improved.

### 1. Clarity

I found really confusing the constant back and forth between
- VIPS and iBayesGMM
- existing work and contributions.

For example:
- In Section 1:
*“Namely, VIPS uses samples from the individual components to estimate the natural gradient using the policy search method MORE (Abdolmaleki et al., 2015), which is also applicable to non-differentiable target … outperforms both previous methods significantly"*.

- In Section 2:

(a) In Section 2.2.1: Important aspects regarding iBayesGMM updates are deferred to Section 3.1.2, blurring the line between existing work and contributions. In my view Section 2.2.1 should be devoted to summarizing the key aspects of iBayesGMM, including theoretical guarantees, parameters updates and implementation details.

(b) Similarly, I think that Section 2.2.2 should be devoted to summarizing the key aspects of VIPS. Please also refrain from using “us” when you are referring to previous existing work (e.g. “prevents us”).

(c) Section 2.2.3 should be in a separate section to clearly indicate that it is a contribution. Please state your result in a proposition/theorem environment and write down formally the proof in a proof environment. Please also write down the proof that the gradient of the KL divergence between two equal distributions is zero. Can you explain the second inequality after “Furthermore, the gradients of the entropy terms are equal, since”? In addition, the authors should comment more on the implications of their result (theoretically and from an implementation perspective).

 - In Section 3: I found it difficult to differentiate between related work and the authors’ contribution here. Algorithm 1 appears to present a unified framework, yet most of Section 3 seems like related work?
Consider using different names for each option instead of Option 1 and Option 2 everywhere (e.g. Option A1, A2, B1, B2, …)

NB: It might help to have a table summarizing the differences between VIPS and iBayesGMM once these two algorithms have been introduced in lieu of the paragraph from Section 1 mentioned above (from both a theoretical and an implementation point of view).

### 2. Related work
- I think the authors should consider acknowledging recent works of Variational Inference using Mixture Models, such as:

[1] Warren Morningstar, Sharad Vikram, Cusuh Ham, Andrew Gallagher, Joshua Dillon. Automatic Differentiation Variational Inference with Mixtures. Proceedings of The 24th International Conference on Artificial Intelligence and Statistics, PMLR 130:3250-3258, 2021.

[2] Kamélia Daudel, Randal Douc, and François Roueff. Monotonic alpha-divergence minimisation for variational inference. Journal of Machine Learning Research, 24(62):1–76, 2023.

- I was also surprised not to see a reference to:

[3] Samuel Gershman, Matt Hoffman, and David Blei. Nonparametric variational inference. In Proceedings of the 29 th International Conference on Machine Learning, Edinburgh, Scotland, UK, 2012.

- *“Arguably the two most effective algorithms for GMM-based VI”*: Perhaps the authors could make a less assertive statement here, in light of new bodies of research for GMM-based VI such as [1,2].

### 3. Minor comments
- *“Their approach can be shown to converge, even when the M-Step does not consist of single natural gradient updates, however, it was not yet proven, that their proposed procedure, which does use single natural gradient steps, also performs natural gradient descent on the full mixture.”*
Consider splitting this sentence in two.

- *“two equal distribution is zero”*
Typo: an s is missing

- *“Pseudo-Code for our framework is show in Algorithm 1.”*
Typo: shown

- A parenthesis is missing at the end of eq. (4) and in the definition of R(o) after eq. (11)

---

> ### Author Response · Authors · 2023-05-26
> **Thank you for your helpful comments**
>
> Thank you very much for your valuable feedback regarding the structure of our paper.
> If we understand correctly, you recommend a structure roughly as follows:
>
> 2.1.1 Theory iBayesGMM
>
> 2.1.2a IBayesGMM Design Choices A
>
> 2.1.2b IBayesGMM Design Choices B
>
> ...
>
> 2.2.1 Theory VIPS
>
> 2.2.2a VIPS Design Choices A
>
> 2.2.2b VIPS Design Choices B
>
> ...
>
> 3.1 Unification of VIPS & IBayesGMM
>
> 3.2 Our Generalization
>
> We agree, that such structure would reduce the back-and-force between iBayesGMM, VIPS and our contribution. We also considered such structure initially, but we decided against it for the following reasons: There would be substantial back-and-forth between theory and the different implementation details. For example, for our unification of VIPS and IBayesGMM, the only relevant sections (2.1.1, 2.2.2, 3.1) would be spread far apart. The reader will likely not recall all relevant aspects/equations, so they would need to scroll between the different section. Furthermore, they would have learned at that point a lot about different implementation details that are actually not relevant for showing that the derived update equations are equivalent, which could be distracting. Similarly, the design choices for each of the different options (and their implementations in our modular framework) would also be spread far apart. For example, the different design choices for Option A would be discussed in 2.1.2a, 2.2.2a and 3.2, which would make it much more difficult for the reader to grasp the differences.
>
> Instead, we opted for strict separation between theory and design choices, where in the first part of the paper we contrast the different derivations and show that they actually result in the same update rules; and in the second part, we focus on the different options, one after each other. We argue that this separation is well aligned with the main point of our work: That the derived update equations are the same, but the respective implementations are quite different.
> Furthermore, we opted for side-by-side comparisons consistently throughout the paper to make it convenient to the compare the different approaches (iBayesGMM, VIPS, our generalization) - first in their derived update equations, then in terms of the implementations. This originally led to the following structure:
>
> 2.2 Theory
>
> 2.2.1 iBayesGMM
>
> 2.2.2 VIPS
>
> 2.2.3 Our Generalization
>
> 3. Implementation Details
>
> 3.1 Option A (iBayesGMM vs VIPS vs our)
>
> 3.2 Option B (iBayesGMM vs VIPS vs our)
>
> While we would like to stay close to our proposed structure for the reasons mentioned above, we updated our presentation significantly, which hopefully also addresses the issues that you raised.
>
>  1) We realized that our structure could be confusing to a reader that expects the paper to present the full approaches---including implementation details---one after the other. We addressed this by explaining the upcoming structure at the end of the introduction, as this will give the reader a better guidance.
>
> 2) To further emphasize to the reader that we opted throughout the paper for side-by-side comparisons between iBayesGMM, VIPS and our unification we improved the structure and formatting of the sections where we discuss the different design choices. Namely, each of these subsections now contains a paragraph for iBayesGMM and one for VIPS, which we indicate using bold fonts. Furthermore, each subsection contains a textbox where we explain, how these different options are realized in our framework / implementation. We believe, that this formatting and structure would also help a reader who wants to better understand the design choices of one particular method, as it is now very convenient to identify the relevant paragraphs and to ignore the irrelevant ones, despite the fact, they are spread out a bit due to our interleaved presentation.
>
> 3) We also realized that it can be difficult to identify which sections discuss prior work and which sections discuss our theoretical unification and the resulting implementation. We addressed this by combining the list of our contributions with the explanation of our structure, telling the reader exactly at which sections they can expect novel insights. We also believe that adding distinct paragraphs and the textboxes helps in better understanding the differences in the implementations.
>
> 4) Following your suggestions, we state our theoretical result now in the form of a theorem, and we added a discussion of these results to Section 2.3. We had to move the proof itself to Appendix D. The upside of this is, that we could elaborate more on the proof which should make it easier to follow. We also added a lemma with a proof that the gradient of the KL divergence between equal distributions is zero. To address your question about the second equality: The equality follows from the facts that the integral over $q_{\boldsymbol{\theta}}(\mathbf{x})$ is one and that $q_{\boldsymbol{\theta}}(\mathbf{x}) = \tilde{q}(\mathbf{x})$ directly after the E-Step.

---

> > ### Comment · Reviewer_dU7d · 2023-06-12
> > **Thank you for you reply**
> >
> > I want to thank the authors for their reply and for their efforts in improving their work. I have a few more comments which need to be addressed.
> >
> > ### Theorem 2.1
> > The statement and proof of Theorem 2.1 can be improved. In particular, the authors should write more clearly in the main text the equation that they prove, which I think is: for all component parameters $\theta, \theta’$, \frac{\partial}{\partial \theta} \tilde{J}(q_{\theta’}, \theta)|_{\theta = \theta’} = \nabla J(\theta)|_{\theta = \theta’}.
> >
> > Then they can explain that the E-step corresponds to setting $\tilde q = q_{\theta^{(i)}}$, where $\theta^{(i)}$ denotes the parameters at time $i$, so that
> > - the objectives match after the E-step (i.e. \tilde{J}(q_{\theta^{(i)}}, \theta^{(i)}}) = J(\theta^{(i)}}))
> > - their gradients match as well after the E-step by directly applying Theorem 2.1 (\frac{\partial}{\partial \theta} \tilde{J}(q_{\theta^{(i)}}}, \theta)|_{\theta = \theta^{(i)}}} = \nabla J(\theta)|_{\theta = \theta^{(i)}}}).
> >
> > In the proof, please write |_{\theta = \theta’} instead of |_{q_\theta = \tilde q} and replace \partial/\partial \theta by \nabla where needed.
> >
> > ### Some other minor comments:
> > - We find both the notation Sec. 2.1 and Section 2.3 in the manuscript, I would suggest only using the latter throughout the paper.
> > - In the equality following equation (2), I believe it should read q_\theta(o|x) in the KL term and similarly in the following sentence “that is tight when … = q_\theta(o|x).”
> > - The citations for Morningstar et al. (2021) and Daudel et al. (2023) have been conflated in Appendix C

---

> > > ### Author Response · Authors · 2023-06-13
> > > **Thank you for the additional feedback**
> > >
> > > Thank you very much for the engagement. We further improved our presentation as follows:
> > >
> > > 1. We noticed that our use of $\frac{\partial}{\partial \boldsymbol{\theta}}$ in the appendix was not consistent with our notation in the main document, where we used $\nabla_{\mathbf{y}}$ to refer to the (partial) gradient with respect to a vector $\mathbf{y}$. Hence, we now always use $\nabla_{\mathbf{y}}$ instead of $\frac{\partial}{\partial \mathbf{y}}$. We also use this subscript notation when the function only depends on a single variable, that is, we use $\nabla_{\boldsymbol{\theta}} J(\boldsymbol{\theta})$ instead of $\nabla J(\boldsymbol{\theta})$ to make the equations slightly more explicit.
> > >
> > > 2. We now use $\tilde{J}(q_{\boldsymbol{\theta}^{(i)}}, \boldsymbol{\theta})$ instead of $\tilde{J}(\tilde{q}, \boldsymbol{\theta})$ when referring to the lower bound directly after the E-Step (where $\tilde{q} = q_{\boldsymbol{\theta}^{(i)}}$). We use this term in Section 2.2 to more explicitly explain how the E-step is performed, and we also use it now in Theorem 2.1 for showing the equation that we prove:
> > > $\tilde{\nabla}\_{\boldsymbol{\theta}} J(\boldsymbol{\theta})\Big|_{\boldsymbol{\theta}=\boldsymbol{\theta}^{(i)}}  = \tilde{\nabla}\_{\boldsymbol{\theta}} \tilde{J}(q\_{\boldsymbol{\theta}^{(i)}}, \boldsymbol{\theta}) \Big|\_{\boldsymbol{\theta}=\boldsymbol{\theta}^{(i)}}$.
> > > Furthermore, this notation now allowed us to use $\Big|\_{\boldsymbol{\theta}=\boldsymbol{\theta}^{(i)}}$ instead of the slightly inaccurate $\Big|\_{q\_{\boldsymbol{\theta}}=\tilde{q}}$.
> > >
> > > 3. We addressed your minor comments by changing "Sec" to "Section", by changing $q$ to $q_{\boldsymbol{\theta}}$ where appropriate and by swapping the citations for Morningstar et al. (2021) and Daudel et al. (2023).

---

> ### Author Response · Authors · 2023-05-26
> **Related Work and Minor Comments**
>
> We extended our discussion of related work by now also discussing the papers you mentioned. Thank you very much for bringing them to our attention! However, please note that [1] tackles amortized variational inference, which is different from our problem setting, since we want to directly optimize the parameters of the GMM. Furthermore, [2] consider alpha-divergences with $0 \le \alpha < 1$, and, therefore, can not be used for minimizing the KL-divergence, which is obtained for $\alpha=1$. Finally, while NPVI technically optimizes a GMM, they only consider a very limited special case, with Gaussian components with isotropic covariance matrices and uniform weights. We changed the sentence in the introduction to "Arguably the two most effective algorithms for maximizing Equation 1," to clarify that we are focusing on the problem setting of non-amortized VI, with full covariance GMMs in the (typical) reverse KL formulation.
>
> We addressed your minor comments. Thank you also for bringing these to our attention

---

### Review · Reviewer_fDiF · 2023-05-26

**Summary Of Contributions:**

This paper presents a unified perspective of variational inference for Gaussian Mixture Model (GMM) which combines two lines of independent work, including VIPS and iBayes-GMM.

In particular, it is shown that VIPS further relaxes the vanilla VI's ELBO of GMM with another lower-bound, which is interestingly optimized via an EM procedure. It is then claimed (by this paper) that the gap between the ELBO and its derived lower-bound in VIPS would zero out (with respect to the current estimate model parameter) after every expectation step (E-step).

Thus, at the beginning of each maximization step, with the above gap dismissed, it follows that per iteration, the objective of M-step in VIPS is the same with the objective in iBayes-GMM.

With this unified view, it follows naturally that the specific implementation subroutines (i.e., approximation to exact optimization or computation of intermediate quantities) used in each approach (i.e., VIPS & iBayes-GMM) can now be mixed and matched, resulting in a broader implementation scheme comprising 432 combinations of those implementation subroutines.

**Audience:**

Yes

**Claims And Evidence:**

No

**Requested Changes:**

I request the following changes:

1. Elaborate on how the gap between the ELBO & VIPS's relaxed version disappear. I believe this is key to establishing equivalence. Otherwise, if this is assumed away, the rest of the equivalence might be a bit too straightforward to make a contribution.

2. Adding comprehensive coverage for VIPS, iBayes-GMM along with background on natural gradient & Fisher Information Matrix.

3. Empirically, please demonstrate that the VIPS's ELBO gap is tight after each E-step as claimed.

**Strengths And Weaknesses:**

STRENGTH:

This work is highly practical and has an impressive amount of empirical work.

The paper is also very well-written and has a clever way of presenting the material in such a way that the key idea & its most important execution would come across clearly even if the reader are not familiar with the prior work of VIPS and iBayes-GMM.

The key idea is also pretty intriguing to me: by forging a mapping between the high-level layout of the two approaches, one can then mix and match their specific approximation subroutines (originally devised to each component of the high-level map)

WEAKNESS:

Despite the strength above, I unfortunately am not quite convinced with the theoretical component of this paper, which might be both incomplete and flawed. Let me elaborate below:

First, the established equivalence between VIPS & iBayes-GMM appears to rest of the claim that the gap between the vanilla ELBO and the relaxed version of VIPS would disappear after each expectation step in the EM scheme of VIPS.

Given that the gap disappears, the equivalence would obviously follow but the claim is assumed rather than being proven formally. This kind of makes the claimed equivalence vacuous.

Second, another key step in establishing this equivalence lies with the equation following Eq. (5) in which it is shown that the gradients of the entropy terms are equal. But, the derivation does not seem right to me. Please correct me if I miss any essential steps below:

From the presented derivation, I get that:

$\frac{\partial}{\partial\boldsymbol{\theta}}\Bigg[\displaystyle\int_{\mathbf{x}} q_{\boldsymbol{\theta}}(\mathbf{x})\log q_{\boldsymbol{\theta}}(\mathbf{x})\mathrm{d}\mathbf{x}\Bigg] = \displaystyle\int_\mathbf{x} \Bigg[\frac{\partial q_{\boldsymbol{\theta}}(\mathbf{x})}{\partial\boldsymbol{\theta}}\cdot \log q_{\boldsymbol{\theta}}(\mathbf{x})\Bigg]\mathrm{d}\mathbf{x}$

Then, using the same logic

$\frac{\partial}{\partial\boldsymbol{\theta}}\Bigg[\displaystyle\int_{\mathbf{x}} q_{\boldsymbol{\theta}}(\mathbf{x})\log \tilde{q}(\mathbf{x})\mathrm{d}\mathbf{x}\Bigg] = \displaystyle\int_\mathbf{x} \Bigg[\frac{\partial q_{\boldsymbol{\theta}}(\mathbf{x})}{\partial\boldsymbol{\theta}}\cdot \log \tilde{q}(\mathbf{x})\Bigg]\mathrm{d}\mathbf{x}$

which would require $q_{\boldsymbol{\theta}}(\mathbf{x}) = \tilde{q}(\mathbf{x})$. But it is not clear how it will be achieved.

--

Further minor points:

Although the presentation is made in a very clever way that (in my opinion) allows people to appreciate the core idea without needing exposure to previous work of VIPS and iBayes-GMM, I still think that for a journal, it is best that those prior work are covered comprehensively in separated sections to improve the educational value of this work.

It is also good to include detailed background on natural gradient, exponential family, Fisher Information Matrix and etc. A succinct review of the work of (Amari, 1998) would be nice to included (in the appendix if not in the main text)

In addition, it seems Eq. (3) might have missed $q_{\boldsymbol{\theta}}(o)$. Otherwise, how would it be optimized?

---

> ### Author Response · Authors · 2023-05-26
> **Thank you for the constructive feedback**
>
> Thank you for recognizing our empirical work! We have indeed dedicated a great deal of effort to achieving a fair, transparent, and thorough evaluation and a highly usable and performant implementation.
>
> Regarding your concerns about the correctness of our results. Directly after the E-Step the lower bound is indeed tight, since $\tilde{q}(\mathbf{x})$ will be equal to $q_{\boldsymbol{\theta}^{(i)}}$, (where $\boldsymbol{\theta}^{(i)}$ shall denote the current model parameters at iteration $i$). Note that this equality is true by construction: The E-step can be performed simply by copying the parameters of the current model for constructing $\tilde{q}$. However, while $\tilde{J}(\tilde{q}, \boldsymbol{\theta}^{(i)}) = J(\boldsymbol{\theta}^{(i)})$, both objectives will be different for $\boldsymbol{\theta} \neq \boldsymbol{\theta}^{(i)}$ and will have different optima. In our opinion, it is not obvious that the gradients of both objectives match directly after the E-step, because the fact that both losses share the same value at a single point $\boldsymbol{\theta}=\boldsymbol{\theta}^{(i)}$ does in general not imply that also the gradients match at this point.
> Showing that this equality indeed holds, which implies that the natural gradients (only directly after the E-Step) match, and therefore that both algorithms perform the same updates is the main theoretical insight of our work. We have elaborated our proof (which can now be found in Appendix D) to hopefully make it clearer. Please let us know whether the new proof is now convincing.
>
> Regarding your question on Eq.(3): This equation is only used for independently updating a given component $q_{\boldsymbol{\theta}}(\mathbf{x}|o)$. Eq. 3 is obtained from the lower bound by removing constant offsets (the loss of the other components) and a constant factor (the weight of the component that is currently updated).
>
> We have improved our presentation to make it easier to understand both algorithms, despite the fact that different aspects are spread over different sections. We achieved this, by making it easier to quickly identify the relevant paragraphs using bold fonts. We can also devote additional sections in the appendix for both algorithms. However, we probably would not add new content compared to the descriptions that can already be found in the main manuscript. For those details of prior works that are not covered in our manuscript (for example rigorous proofs, or the specific way VIPS uses to add new components) it makes more sense to consult the original works. However, we agree that adding background on natural gradient descent would be topical and potentially helpful, and we will still add this in a revision. In the meantime, please let us know if the current revision satisfactorily addresses the remaining issues.

---

> ### Author Response · Authors · 2023-06-07
> **Added background on Natural Gradient, Fisher Information Matrix and exponential family distributions**
>
> We have just added a section for providing background on natural gradient descent, Fisher information and exponential family distributions (Appendix M).
>
> Please let us know, if you have any further suggestions.

---

### Review · Reviewer_qBLy · 2023-05-28

**Summary Of Contributions:**

This paper shows the equivalence between two GMM-based variational inference methods, VIPS and iBayes-GMM. Based on this new insight, the authors further summarize the design components and design choices in GMM-based variational inference and test these combinations on Bayesian logistic regression on the Breast Cancer dataset, a small Bayesian neural network on the Wine dataset.

**Audience:**

Yes

**Broader Impact Concerns:**

No broader impact concerns.

**Claims And Evidence:**

Yes

**Requested Changes:**

See Weaknesses.

**Strengths And Weaknesses:**

Strengths:

- The studied problem is important and the motivation is clear.
- The provided new perspective appears technically sound
- The comprehensive summary of GMM-based variational inference variants and provided codebase can be very useful for the practitioners

Weaknesses:

- It will be better if the authors can provide a summary of the pros and cons for those combinations and provide some suggestions about which method should be used for which scenarios.
- The evaluation metric is mostly ELBO which sometimes may not align with the metrics that we care about in the end. It will be better to also include other metrics such as MSE or classification accuracy.

---

> ### Author Response · Authors · 2023-06-07
> **Thank you for the helpful comments**
>
> Thank you for recognizing the importance and technical soundness of our work, as well as the relevance for the practitioner.
>
> 1) Regarding the evaluation metric. We evaluated every test problem on a second metric (Table 8 in the appendix).
>
> - Most of the experiments are not based on a data set; instead, the target distribution is given in analytic form, and therefore we can only evaluate how accurately the target distribution is matched. Here, the ELBO is an important metric as it directly corresponds to the Kullback-Leibler divergence that we wish to minimize. However, it is well known, that the reverse KL divergence does not strongly incentivize covering many modes of the target distribution. Hence, we used a second metric that puts more focus on covering many modes. For the GMM and STM toy problems, the modes of the target distributions are known to us, and thus, we can directly evaluate which modes are covered by the approximation. On the PlanarRobot experiment, we follow Arenz et al. (2020) and use the maximum mean discrepancy as second method. For the TALOS experiment, we directly evaluate the entropy of the learned model, to compare the variability of the learned joint configurations.
>
> - We agree that it is interesting to evaluate the posterior predictions for those experiments that are based on a data sets. For this reason, we already opted for the MSE on the WINE experiments (which we introduced in this work), and we found that the results were consistent with ELBO evaluations. However, on the logistic regression experiments, we originally did not evaluate with respect to the classification performance, but focused on the quality of the approximation by using the MMD as second metric. We argue, that---while interesting---, the prediction performance of the approximated posterior is actually not that relevant for our work, which focuses on the problem of learning highly accurate approximations.  If a worse approximation of the posterior achieved better prediction performance, then this would be primarily a problem in the specification of the posterior distribution and not in the approximation method. We also note that the logistic regression experiments were originally introduced by Nishihara et al. (2014) and later used by Arenz et. al. (2020) and in both cases were not evaluated with respect to the prediction performance. However, as we do think, that it is still interesting to evaluated the prediction performance, we reran all logistic regression experiments and evaluated the accuracy of the predictions based on the learned posterior distribution. We added the results to the current revision (Table 9). However, we could not observe significant differences in the prediction performance of the different methods/variants.
>
> 2) Regarding the discussion of Pros and Cons for the practitioner.
> Thank you very much for this suggestions. We agree that such discussion is very useful for anyone who wants to apply our framework in practice. We added a discussion (Appendix M) to the revision, that should help the practitioner in choosing appropriate design choices and hyperparameters.
>
> References:
> ----------------
> Nishihara, R., Murray, I., and Adams, R. P. Parallel MCMC with Generalized Elliptical Slice Sampling. Journal of Machine Learning Research, 15(1):2087–2112, January 2014.

---

### Author Response · Authors · 2023-05-26
**Revision**

We want to thank the reviewers for the constructive comments and for recognizing our theoretical and empirical contributions.

We just updated a revision to address most of the comments, namely:

1. The introduction now already mentions the specific problem setting that we address in our work. Furthermore, the introduction now outlines the structure of the paper and connects the sections to the different contributions.

2. The presentation was improved to make it much easier to find the relevant sections for the different algorithms reviewed in this work.

3. The main theoretical insight is now stated as a theorem, and the implications of the results are discussed. The proof for this theorem can now be found in the appendix and should now be clearer.

4. We extended our discussion of related work.

5. We addressed minor comments and fixed a few typos

We still plan to add background material on natural gradient descent and Fisher information and exponential family distributions to the appendix as requested by reviewer fDiF.

Please let us know, if you have further comments or questions, or if we did not satisfactorily address any of the previously raised issues.

---

### Author Response · Authors · 2023-06-23
**visbility of reply**

We just realized that one of our replies ( https://openreview.net/forum?id=tLBjsX4tjs&noteId=0gwU7xxn5T ) might not have have been visible to some reviewers and fixed the visibilities.

We kindly ask you to excuse this mistake.

---

### Decision · Action_Editors · 2023-07-11

**Recommendation:** Accept as is

**Comment:**

The paper has already undergone three revisions during the reviewing process, improving clarity and addressing reviewers' comments. My understanding is that most, if not all, of the reviewers' suggestions have been incorporated. Therefore, I don't see a need for a further revision, and I recommend acceptance as is.

**Audience:**

The paper is a theoretical contribution on natural-gradient variational inference with Gaussian mixture models followed by an empirical investigation. These findings are of interest to at least some of TMLR's audience.

**Claims And Evidence:**

In their final recommendations, all three reviewers report that the claims are supported by sufficient evidence.

In their initial review, Reviewer fDiF raised some concerns about the validity of the paper's theoretical component, but with further clarification from the authors this concern seems to have been adequately addressed.

Reviewer dU7d requested improvement in the clarity of the manuscript, which seem to have been addressed in the revised version.